# Transmission interference microscopy of anterior human eye

Samer Alhaddad[1,2], Wajdene Ghouali[3], Christophe Baudouin[3], Albert Claude Boccara[1] & Viacheslav Mazlin [1] ✉

Cellular imaging of the human anterior eye is critical for understanding complex ophthalmic diseases, yet current techniques are constrained by a limited field of view or insufficient contrast. Here, we demonstrate that Ernst Abbe's foundational principles on the interference nature of transmission microscopy can be applied in vivo to the human eye to overcome these limitations. The transmission geometry in the eye is achieved by projecting illumination onto the posterior eye (sclera) and using the back-reflected light as a secondary illumination source for anterior eye structures. Specifically, we show that the tightly localized illumination spot at the sclera functions analogously to a closed condenser aperture in conventional microscopy, significantly enhancing interference contrast. This enables clear visualization of cells and nerves across all corneal layers within an extended 2 mm field of view. Notably, the crystalline lens epithelial cells, fibers, and sutures are also distinctly resolved. In patients, Fuch's endothelial dystrophy - a major ophthalmic disease affecting 300 million people - is highlighted under a transmission contrast, providing complementary information to traditional reflection contrast. Constructed using consumer-grade cameras, the instrument offers a path toward broad adoption for pre-screening and surgical follow-up, as well as for diagnosing corneal infections in low-resource settings, where anterior eye diseases are most prevalent.

Anterior segment, composed of cornea and crystalline lens, ranks among the most commonly operated regions of the human body. Every year, 26 million people worldwide undergo cataract surgery on the crystalline lens[1], and 5 million people receive refractive surgery on the cornea[2]. The demand for such procedures is even higher, with an estimated 65 million people currently awaiting cataract surgery and 124 million people experiencing visual impairment or blindness due to uncorrected refractive errors[3]. Although surgeries have very high success rates of above 90%, the remaining 10% still represent millions of people affected by complications. Post-surgical complications range from dry eye disease (DED)[4] - which can cause life-long ocular discomfort/pain and dependence on artificial tears - to endothelial decompensation necessitating the corneal transplantation[5]. Importantly, many undesired

outcomes can be prevented through pre-surgical diagnosis. For example, a high-resolution specular microscopy (SM) is routinely used in clinics to measure corneal endothelial cell density and quantitatively assess the risk of potential decompensation[6]. As a drawback, SM is narrowly specialized in imaging the endothelial layer and is incapable of visualizing other corneal structures, such as nerves that play a key role in DED.

Over the years, several instruments have been developed to enable high-resolution imaging of the entire cornea. Notably, in vivo confocal microscopy (CM) has become part of hundreds of clinical trials and has identified additional cellular biomarkers for DED, corneal inflammation, infections, dystrophies[7,8]. CM has also established the connection between corneal biomarkers and systemic health

[1]Institut Langevin, ESPCI Paris, Université PSL, CNRS, Paris, France. [2]SharpEye SAS, Gentilly, France. [3]Quinze-Vingts National Ophthalmology Hospital, Paris, France. ✉e-mail: mazlin.slava@gmail.com

conditions, such as diabetes[9] and neurological disorders including Parkinson's disease and multiple sclerosis[10,11]. Despite these advances, CM remains challenging to use in daily clinics due to its requirement for direct physical contact with the patient's eye and its limited field of view (FOV), typically below 0.5 mm. More recently, these limitations have been addressed by a generation of high-resolution optical coherence tomography (OCT) devices that combine cellular resolution with an extended FOV of about 1 mm and offer non-contact imaging at a safe distance from the eye[12–23]. However, the high cost and complexity of such systems – often involving ultra-fast scientific cameras, secondary OCT subsystems, or high-frequency galvo scanners - remain a significant barrier to their clinical transfer.

All the above-mentioned technologies operate in reflection (back-scattering geometry); however, it is also possible to image the human eye in transmission. Since the 1970s, ophthalmologists have been familiar with the slit-lamp retroillumination exam, where the back-scattered light from the posterior eye is used to illuminate the anterior eye structures[24]. This back-scattered light is also familiar to photographers as the 'red eye' effect. While retroillumination is useful for a global overview of the eye, its traditional implementation offers low-resolution and weak contrast, which are insufficient for resolving cellular details. One clever approach to see the cells in transmission was recently proposed by T. Weber and J. Mertz[25], who induced phase-gradient contrast[26] in the cornea and crystalline lens using asymmetric large-area illumination (42° visual angle) back-scattered from the posterior eye.

In this work, we show that the foundational ideas of Ernst Abbe[27] and Fritz Zernike[28] regarding the interference nature of transmission imaging are also relevant to the in vivo microscopy of the human eye. Specifically, we demonstrate that interference contrast can be enhanced by employing symmetric illumination focused onto a small, localized spot (6°) on the posterior eye. As shown below, this localized spot on the retina/sclera acts analogously to a nearly closed condenser aperture (aperture controlling the NA of sample illumination) in conventional microscopy and increases the coherent contrast at the expense of a reduction in resolution/depth-of-field (DOF). We present a significant enhancement in the visibility of corneal and crystalline lens structures in the human eye in vivo. Both healthy subjects and those with pathologies are examined, and the imaging results are compared with established clinical modalities (CM, SM) as well as an emerging research time-domain full-field OCT (TD-FF-OCT) device[29]. Additionally, the effects of pupil size and cataract scattering on transmission imaging are investigated.

## Results

### Physical concept of transmission interference microscopy in the eye

The physical principle of the instrument is illustrated in Fig. 1. The light from the near-infrared (NIR) 850 nm light-emitting diode (LED) is focused on the back-focal plane of the microscope objective. The objective produces a quasi-collimated beam that is directed towards the eye and focused by the optics of the eye (cornea and crystalline lens) onto the retina. The sclera, located immediately behind the retina, scatters the incoming illumination, and a portion of the scattered light returns to illuminate the cornea and crystalline lens from the back. The light passing through those layers in transmission is collected by the camera. Finally, the camera produces an image of a single anterior eye layer that is located in the optical focus of the objective. To capture only the transmitted light and avoid the strong specular reflex from the corneal surface, the illumination and detection are cross-polarized. Here, we rely on multiple scattering in the sclera to randomize the pre-encoded linear polarization of the illumination. The optical components are listed in Methods.

The imaged structures in this transmission microscope appear either with dark or bright contrasts depending on their axial location relative to the optical focus (Fig. 1a–c). This effect has been known in bright field transmission microscopy and is attributed to the interference nature of light[27]. The interference occurs between the light diffracted by the sample and the light transmitted through the sample (zero-order diffraction). Analytically, it can be described by calculating the diffraction integrals at the sample, represented as a diffraction grating[30–33], and at the objective aperture, or simply by considering the Gouy phase shift in the Gaussian beam model[34].

The emergence of interference phenomena in transmission can be modeled through simulations of the wave equation, as presented in Fig. 1e–i. See the simulation details in Methods. We modeled the transmission as a 4F system with two lenses (objective lens and tube lens) that conjugate the sample to the camera plane. For simplicity, a single spherical scatterer in the sample is illuminated from below with a quasi-collimated wave of incoherent light. In the absence of the scatterer, the quasi-collimated wave is simply relayed by the 4F optical system, producing a uniform illumination on the camera. In contrast, when the scatterer is placed near the focal plane of the objective lens, a portion of the quasi-collimated wave close to the scatter is diffracted (diffracted wave), while another portion of the wave located further from the scatterer propagates without diffraction (transmitted wave, zero-order diffraction). Although the two waves propagate within the same optical path they exhibit different angular divergences – the diffracted wave fills a significantly larger portion of the NA of the objective lens. This leads to a spatial separation of the waves in the back-focal plane of the objective lens, where the transmitted wave is focused, and the diffracted wave is collimated (see the zoomed, unaveraged Fig. 1e, h and Supplementary Movie 1). The tube lens recombines both waves on the camera, producing dark or bright interference contrast for the scatterer. The thicker the scatterer and the higher its refractive index relatively to the background, the larger the phase delay between the waves, leading to higher contrast. The sign of the contrast, denoting constructive or destructive interference, can be alternated by moving axially the camera, the scatterer, or the lenses (as shown with the light blue arrows in Fig. 1g, i). For a small scatterer, an axial shift about the DOF is sufficient to change the contrast from the brightest to the darkest (Fig. 1i).

It is useful to highlight the conceptual difference between this transmission approach and traditional reflection-based interferometric methods. In reflection-based systems, the sample and reference beams travel along separate paths and interfere only in the camera plane. In contrast, in transmission, interference occurs directly in the imaging plane, in the immediate vicinity of the scatterer. This local interference pattern is then relayed to the camera plane by the microscope objective and tube lens.

The photo of the prototype and its performance with a target is shown in Fig. 2. When the target was placed at the objective focal plane, minimal light was detected. This is because specular reflections from the target were filtered out by a polarization beam splitter, which was in a cross-polarized configuration relative to the incident light. In a similar way, this configuration also allows the instrument to filter out light reflected from the cornea. Conversely, when standard paper was placed behind the target while maintaining the same optical configuration, the target became visible in transmission. The paper acts as a diffuse scatterer (analogous to the sclera at the back of the eye). When the light passes through the target and strikes the paper, the latter randomizes the polarization of this incident light. A portion of this depolarized light can then travel back, illuminate the target in transmission, and pass through the beam splitter to be detected by the camera.

### Enhancement of interference contrast in the eye

The interference contrast is valuable as it enables label-free staining of transparent ocular tissues that would be invisible otherwise. One limitation of this contrast is that it is inherently weak, owing to the small

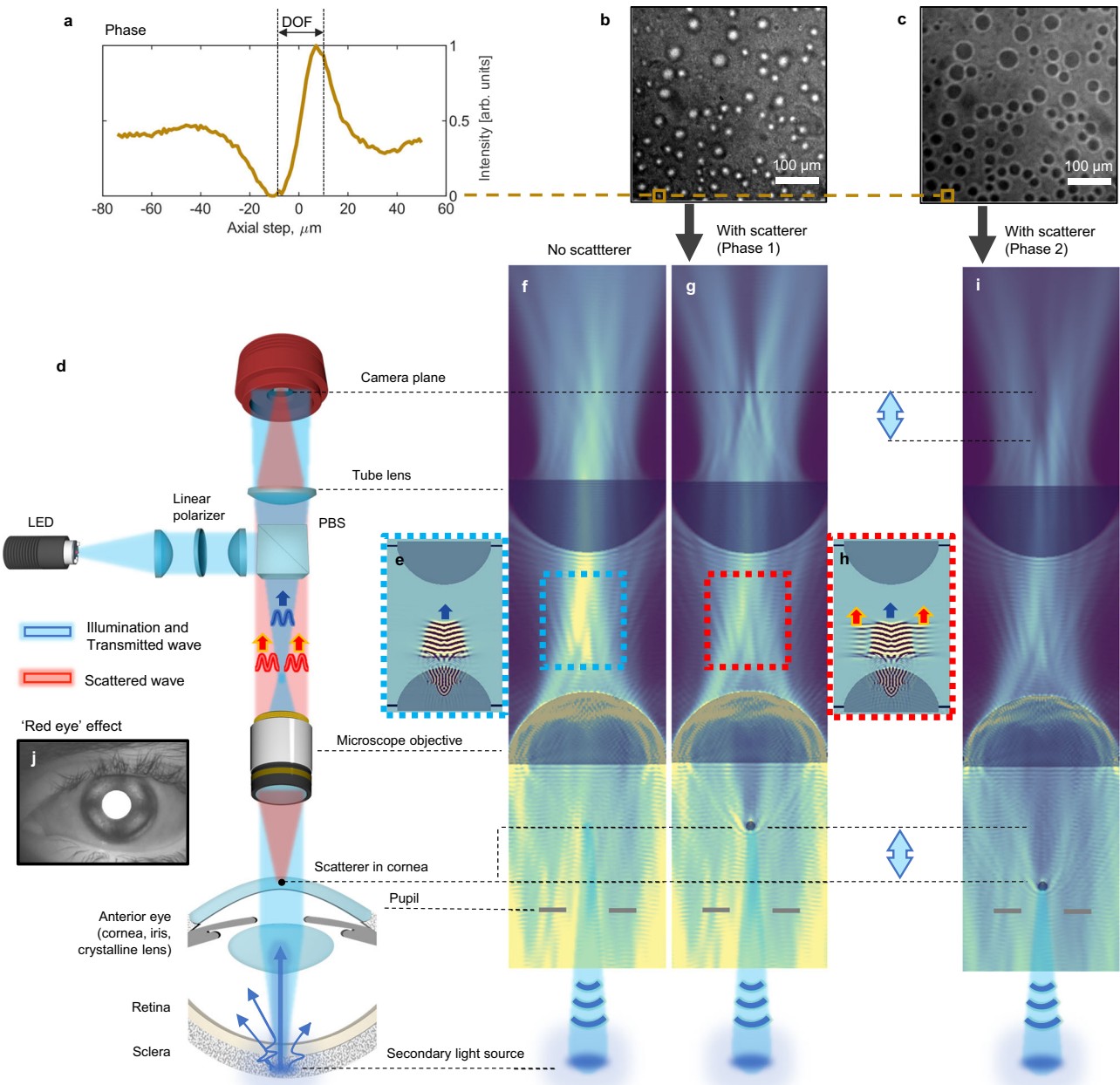

**Fig. 1 | Principle of transmission interference microscopy in the eye.**
**a** Measured intensity around the focal plane. The optical phase exhibits the characteristic π-phase shift. **b**, **c** Transmission images from artificial eye used to compute the curve in a. **d**, Optical scheme of eye imaging device. **f**, **g**, **i** Wave propagation simulations of equivalent 4F transmission imaging in the cases: without the scatter (**f**), and with the scatterer (**g**). The images are produced by time-averaging. **e**, **h** The zoomed regions are produced without time-averaging and illustrate the separation of the focused transmitted wave (blue arrow) and collimated scattered wave (red arrows) in the back focal plane. The movie of coherent wave propagation can be found in Supplementary Movie 1. **i** The axial shift of the scatterer around the focal plane (see the light blue arrows) changes its visible brightness in the camera plane, in agreement with the experiment. **j** The retro-illuminated light from the back of the eye, as seen through the imaging system using small magnification optics instead of the microscope objective.

phase retardation between the transmitted and diffracted waves. Since at least the early 20th century, it has been known that the visibility of scatterers in transmission microscopy can be controlled using the condenser aperture of the source[35]. By narrowing the aperture, the NA of the illumination beam - and thus the spatial coherence - of light illuminating the sample is reduced, which significantly enhances the interference contrast. We show that the same principle holds true for transmission imaging in the eye. In particular, the size of the light spot projected onto the sclera at the back of the eye affects the NA of the secondary source illuminating the anterior eye structures from the back and, consequently, determines the contrast.

A transmission microscope with interchangeable lenses and an aperture in the illumination arm was used to control the magnification of the source image projected onto the sclera (see the concept in Fig. 3a and the full instrument in Fig. 4a). For proper comparison, we imaged the same corneal location of the artificial model eye in all illumination configurations. See the details of the optical configurations in Methods. We observed a drastic increase in interference contrast when the size of the projected source was reduced (Fig. 3b). This tendency remained consistent across the full range of source magnifications tested, from 15 mm to 0.3 mm. On the flip side, the increase in contrast was accompanied by a decrease in DOF, visible as elongation of the axial phase

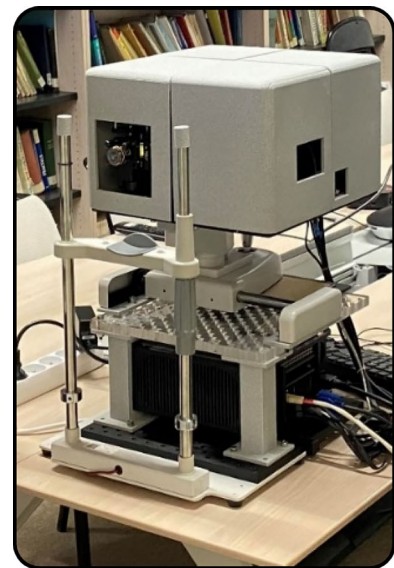

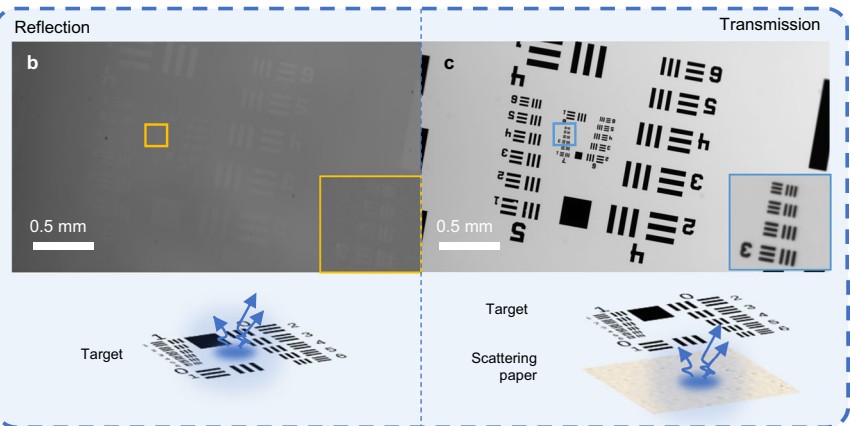

**Fig. 2 | Prototype and transmission performance with the target. a** Tabletop clinical research prototype with a computer. **b**, **c** Illustration of the instrument's capability to suppress reflected light while detecting transmitted signals. A 1951 USAF target (R1DS1P1, Thorlabs, USA) was used as a sample. Both transmission and reflection tests were conducted using the same low optical power (a few mW).

profiles. Importantly, these changes were observed by adapting the illumination only, while the detection path remained intact.

The key idea for understanding these effects is that the source at the sclera (posterior eye) is always located at a distance equal to the length of the eye (about 23 mm) further from the image plane (anterior eye). As a result, the NA of the secondary illumination visible to the scatterer in the anterior eye is determined by the size of the source as: $NA_i = n_{cornea-lens} \cdot \sin[\arctan(\frac{Source\ radius}{Eye\ length})]$, where $n_{cornea-lens}$ is approximated as $1.4$[36]. The $NA_i$ affects both the lateral resolution and DOF, which are commonly determined from the algebraic mean of the $NA_i$ of illumination and $NA_d$ of detection[37]: $\Delta x = \frac{0.61\lambda}{NA_{eff}}$ and $DOF = \frac{n_{cornea-lens} \cdot \lambda}{NA_{eff}^2}$, where $NA_{eff} = \frac{NA_i + NA_d}{2}$. Indeed, experimentally, the scatterers illuminated with a small source (0.3 mm) and low $NA_i$ ($NA_i \approx 0.01$) experience a smaller $NA_{eff}$ and therefore fade at a slower rate with defocus than the same scatterers illuminated with a large source (15 mm) and high $NA_i$ ($NA_i \approx 0.43$) (Fig. 3b). As an additional note, in the latter case the $NA_i$ exceeds the numerical aperture of the microscope objective $NA_d = 0.3$ (used in that experiment); therefore, only the portion of illumination within the acceptance cone of the objective $NA_i = 0.3$ is utilized.

The improvement of contrast with a smaller source size is explained with the simulation in Fig. 3c. When the source is large, the scatterer is exposed to a broad range of illumination angles. Each illumination angle produces a dark-bright phase profile in the camera plane that is tilted relatively to the optical axis. After summing up all these profiles, the contrast is noticeably reduced. Conversely, with a point source, the scatterer is illuminated by light of higher spatial coherence, originating from a smaller illumination angle. This results in a more defined phase relationship between the transmitted and diffracted waves, leading to higher contrast.

In the clinical research prototype the illumination was designed with the intention to enhance the interference contrast over the resolution. The contrast is higher for a smaller illumination spot on the sclera. However, the conservation of *étendue* for an LED source sets the lower bound on the minimal spot size that can be achieved without losing light. Using a microscope objective with 0.3 NA, we produced a

1:1 projection of the LED chip onto the sclera resulting in a small 1.7 mm illumination field. Importantly, one has to also consider the enlargement of the spot size due to the diffuse scattering of the sclera. Using the mean free path of the sclera[38,39] it is possible to estimate the size of the enlarged spot, which in our case is 2.4 mm (see Methods). Using this size, we can calculate the lateral resolution/DOF of 2.8 μm/34 μm. Based on Fig. 3d, one can deduce that this illumination is expected to provide a 3x gain in interference contrast compared to the other known retroillumination microscopes with extended (12 mm = 40°) illuminations[25,40]. We also tested the high-resolution configuration with a microscope objective of 0.4 NA. The 3 mm illumination spot was diffused to about 3.4 mm, resulting in a 2 μm/19 μm lateral resolution/ DOF, respectively.

Here, it is important to note that the formula for the $NA_{eff}$ that we used to estimate resolutions is known for being an approximation[41], and is, in fact, dependent on the scattering within the sample. Indeed, according to Mie theory, the small isolated particles with the size comparable to the wavelength of light exhibit isotropic forward scattering that fills the detection $NA_d$ independently of illumination $NA_i$, so that $NA_{eff} = NA_d$[42,43]. This means that, in transmission, the isolated smallest particles could be seen with a better resolution determined by $NA_d$, independently of $NA_i$. On the other hand, larger particles have predominantly forward scattering that keeps the angular divergence of the illuminating light $NA_{eff} \approx NA_i$. The latter means that the larger particles can hinder viewing across an extended DOF determined by $NA_i$. Finally, the periodicity in the organization of the scatterers also determines the diffraction angles and $NA_{eff}$, similar to the effect seen with a diffraction grating.

## Effects of pupil size and cataract scattering on the interference contrast

Reduced pupil size and increased scattering of the crystalline lens (cataract) are common among a large population of elderly subjects. To study their effects on interference contrast, we used an artificial model eye having a replaceable anterior part featuring either a dilated/ contracted pupil or transparent/scattering lens (Fig. 4).

The pupil size determines the width of the quasi-collimated illumination beam injected into the eye and, therefore, directly affects the

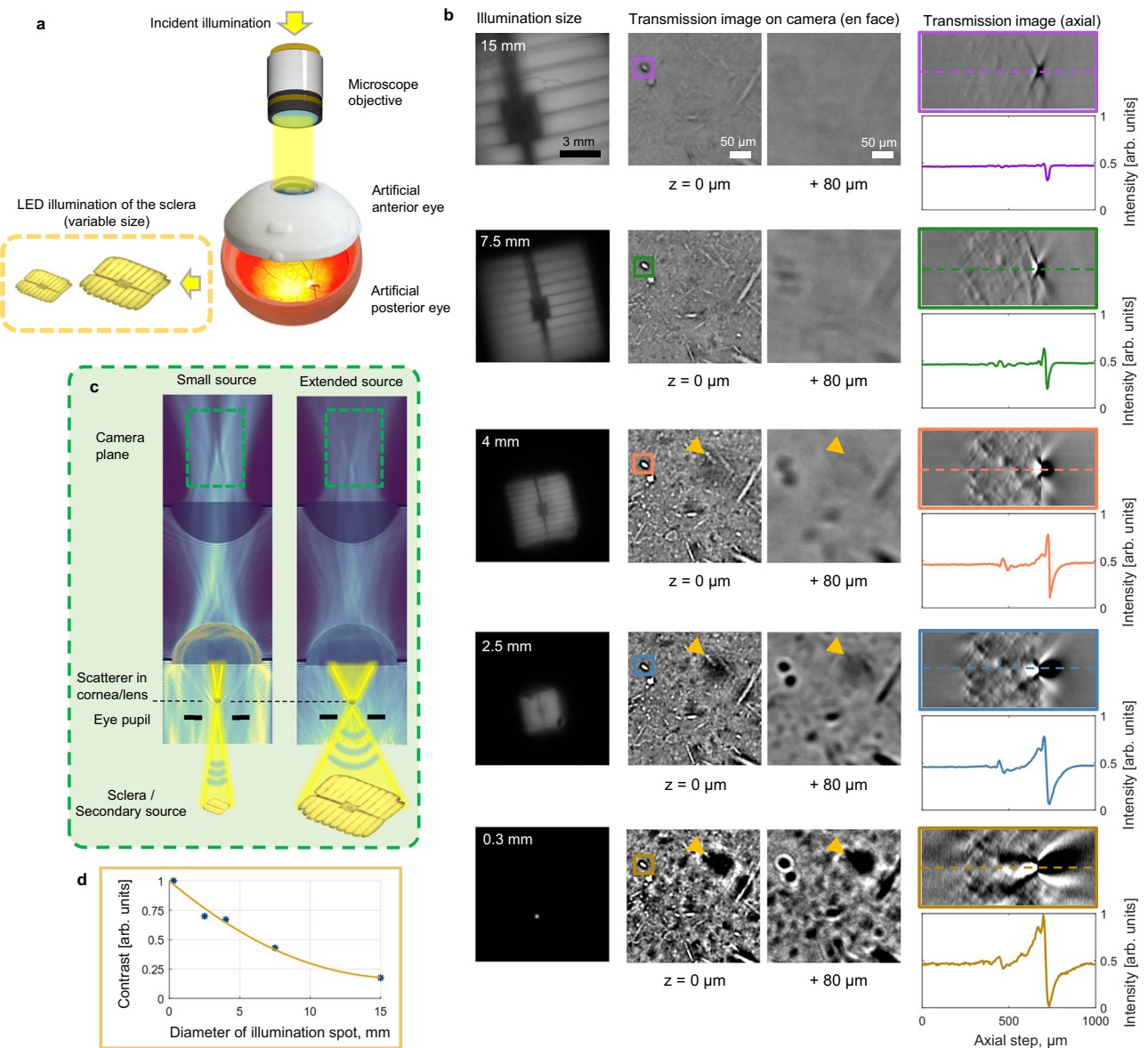

**Fig. 3 | Control of contrast and resolution in transmission interference microscopy. a** Reduced optical diagram of the experiment. The full optical scheme that shows the lenses and iris, used to adapt the size of the projected spot to the back of the artificial eye, can be found in Fig. 4a. **b** Left column: Different projected source sizes, measured experimentally by replacing the posterior part of the artificial eye with the camera. Middle column: En face transmission microscopy images from about 0.5 mm deep inside the cornea (denoted as 0 μm) and from about 0.58 mm deeper layer (denoted as 80 μm). Yellow arrows illustrate faster defocus of the dark spot (smaller DOF), when the illumination spot projected at the back of the eye is smaller. Right column: Axial cross-sectional images produced by stacking the zoomed en face images of the scatter. The 2D profile of the scatter highlights the greater interference phase shift contrast with smaller source sizes. The 3D stacks for different source sizes are visualized in Supplementary Movie 2. **c** Simulation of reduction in contrast and increase in DOF following an increase in the source size at the back of the eye. **d** Quantitative dependency of contrast on the diameter of the illumination spot measured from **b** by calculating the difference between the maximal and minimal values of the axial profile for each spot size.

amount of detected light on the camera (Fig. 4b). Although it is optically feasible to produce a narrow illumination beam that would not be partially blocked by the pupil, this implementation comes at the cost of reduced contrast, either due to increased source magnification or less efficient light collection from the LED. Consequently, in experiments involving human subjects, we opted for a more direct solution: imaging in a dark environment to promote natural pupil dilation and maximize effective illumination.

The scattering of the crystalline lens causes the illumination to diffuse across a larger area of the sclera, thereby increasing the $NA_i$ of the secondary illumination. This leads to a reduction in interference contrast but also results in a shallower DOF, which improves resolution (Fig. 4c). A similar effect is also produced by high ocular aberrations.

## In vivo imaging of human subjects

For deployment in clinics, we constructed a compact (30 cm × 30 cm × 40 cm) tabletop prototype (Fig. 2a). It included the optical setup shown in Fig. 1d, mounted on a manual joystick stage for XYZ alignment to the eye, as well as the electronic controllers and a mini-PC. We tested two budget astrophotography cameras, both with high quantum efficiency in NIR: ASI432 (ZWO, China) with a global shutter for undistorted high-resolution imaging and ASI585MC (ZWO, China) with an increased number of pixels, allowing for extended FOV. See the full list of equipment with specifications in Methods.

Four healthy subjects and two subjects with anterior eye abnormalities were imaged. The images directly contained interference signals, and only minimal processing was required. For

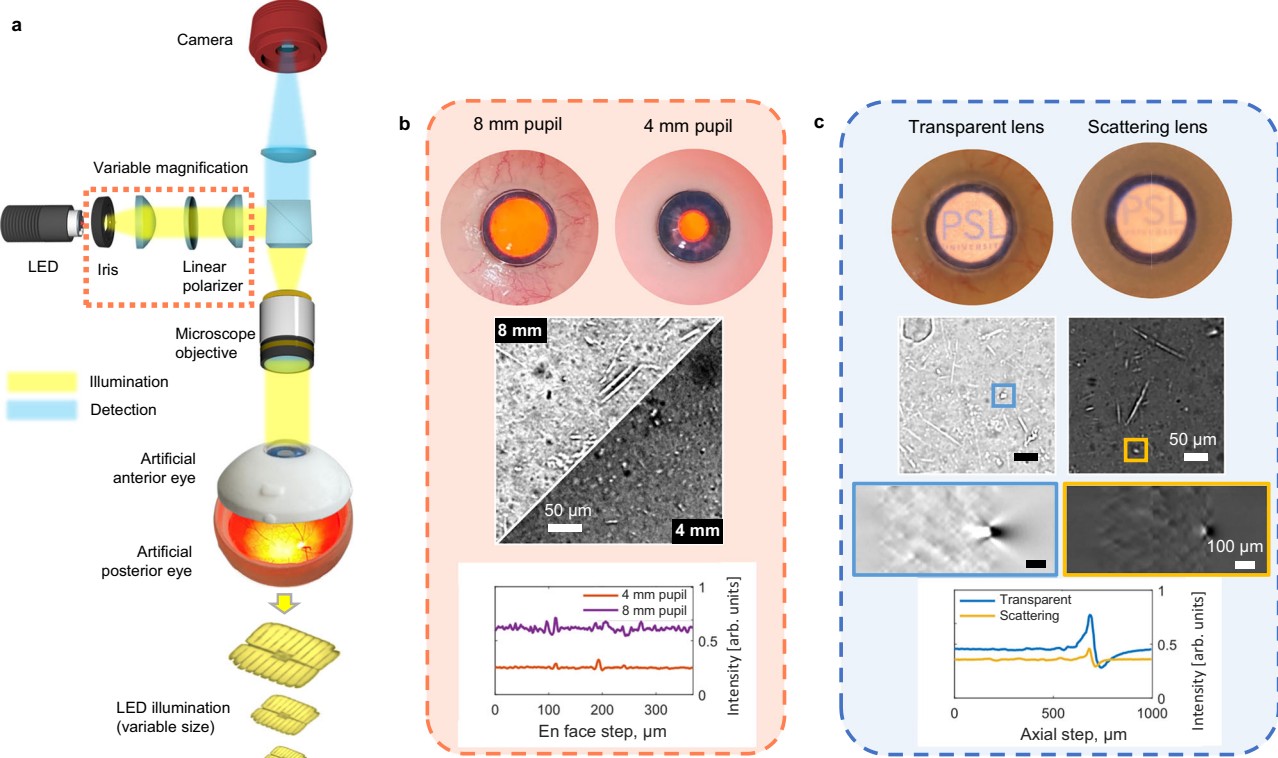

**Fig. 4 | Impacts of the pupil size and cataract scattering on transmission interference signal. a** Full device from the experiments with the artificial eye. **b** Effect of pupil size on the signal. **c** Influence of crystalline lens scattering (cataract) on the interference contrast and on DOF/resolution. The strong scattering of the lens is evident in the macroscopic view through the artificial eye, where the letters behind the lens become unreadable due to pronounced diffusive scattering.

instance, the processing of images captured with the ASI585MC camera was limited to high-pass filtering to remove the uniform light background.

Corneal epithelial cells were visible across an extended 1 mm FOV (Fig. 5a). We estimated the signal-to-noise ratio of 35 dB in Fig. 5a by comparing the signal in the cornea with the standard deviation of noise in the corner of the image, outside of the eye. Despite the supposedly limited axial resolution, we could distinctly identify the superficial, wing, and basal types of cells located at different layers within the 50 μm thick epithelium (Fig. 5b, c). The average cell diameters of the corresponding layers measured 40 μm, 20 μm and 10 μm, respectively. By implementing an additional focus-tunable lens in the detection, it was possible to acquire a fly-though volume of the epithelium (see Supplementary Movie 3 and Methods).

Immediately below the epithelium, we observed the sub-basal nerves as well as the dendritic cells (Fig. 5d). Nerve density biomarker could be measured by segmenting nerves using a custom U-net neural network (NN) trained on our transmission data. A segmented image with the full FOV is provided as Supplementary Fig. 1 and the NN architecture is mentioned in Methods. The measured density of 22 mm/mm² was in agreement with the state-of-the-art CM results from the same subject (different corneal location) (Fig. 5f), as well as with values reported in the literature[44]. The implementation of neural networks for segmentation is particularly valuable in the context of future clinical trials, where the combination of large amounts of data and the extended field-of-view provided by the transmission method would render manual segmentation impractical.

Half-millimeter deep inside the cornea, we captured endothelial cell layer (Fig. 6a). The cells were visible with bright or dark interference contrast, depending on their axial location relative to the optical focus. Due to the natural corneal curvature, both contrasts could be observed within the same image. Importantly, in the exact optical focus, the phase difference between the transmitted and diffracted waves is minimal, rendering the cells invisible. To avoid ambiguity, cell density was calculated by counting only the bright cells and dividing their number by the area in which they were observed. The measured density was 3200 cells/mm², which was within the healthy range according to the literature[45].

The first clinical patient was a 78-year-old male diagnosed with Fuch's endothelial dystrophy (FED). This patient also had a clinically confirmed cataract and was enlisted for provisional cataract surgery. The characteristic guttae of FED were visible (Fig. 6d). Moreover, the guttae exhibited dark-bright shifts across the focus further confirming that they are primarily 'phase' and not light-absorbing objects. In comparison, reflection-based SM depicts guttae as dark spots (Fig. 6e). Moreover, the transmission design gives access to 2 mm×1 mm FOV, which is about 5× larger than the SM (0.25 mm × 0.55 mm)[46] and CM (0.4 mm × 0.4 mm)[47] in clinics. That being said, the reflection methods are fundamentally better at highlighting the cell edges, which are important clinical biomarkers. Indeed, cell edges are typically tilted compared to the cell facets and can redirect the light outside of the detection $NA_d$, thus making the edges seen as dark in reflection. In contrast, the transmission is not sensitive to the tilt angle but to optical thickness. Since the optical thickness is minimal at the edges, the phase shift is small, and the transmission contrast is weak. On the other hand, the nuclei and the main cell body are thicker and give a strong signal in transmission.

Corneal stroma appears in transmission as relatively uniform (Fig. 7a) - a striking divergence from its appearance in reflection, where bright keratocyte cells are highlighted over the dark background (Fig. 7b). We hypothesize this difference to be caused by the unique organization of the stromal fibrils that is responsible for corneal transparency. Indeed, previous simulations have shown that mutual interference between light waves scattered by the fibrils preserves the

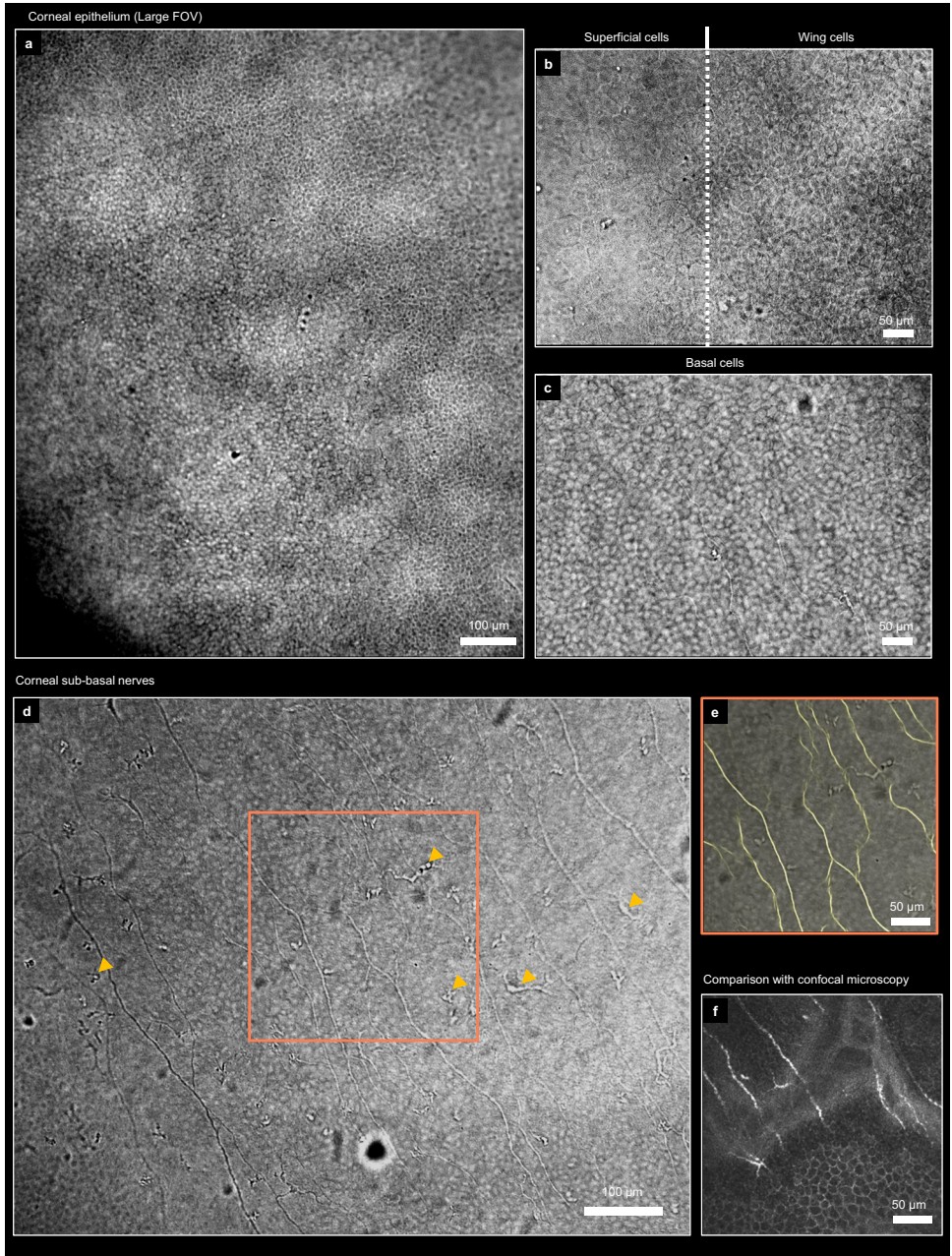

**Fig. 5 | Anterior cornea of healthy human subjects in vivo. a** Corneal epithelium imaged with a large FOV captured in a single camera frame using 0.3 NA objective and ASI585 camera. **b, c** Detailed look at the superficial, wing and basal epithelial layers provided by the 0.4 NA objective and ASI432 camera. The fly-though volume of the epithelium is provided in Supplementary Movie 3. **d** Sub-basal nerve plexus with visible nerves, basal cells and dendritic cells (yellow arrows). **e** Overlay of nerve image with NN segmentation. The full segmented FOV is provided as Supplementary Fig. 1. Even during short sub-second acquisition the nerves alternate their phase and their visible intensity between dark and bright (see Supplementary Movie 4) due to the presence of the fast axial movements, known for the cornea[68]. **f** Comparison with confocal microscopy image obtained on the same subject (different corneal location). Finding the same corneal location was notably difficult with CM due to its small field-of-view and contact nature, where additional pressure on the cornea can cause the layers to curve, further reducing the useful imaging field. Consequently, the results were strongly dependent on the clinical operator's expertise.

wavefront[48]. In our experiment, this implies that the diffracted wave has the same phase as the transmitted wave, and, therefore, no interference contrast can be observed. Although only a few structures are visible in the healthy stroma, this may not hold true in pathological cases. For example, in an elderly 82-year-old male subject, the images revealed fiber bundles passing through the stroma corresponding to corneal microfolds (striae) related to the aging process. In terms of semiology, such striae are interesting, as they could also be an important biomarker for the early detection of keratoconus disease[49], which is characterized by the mechanical deformation of the cornea.

Another strategy to visualize the stromal keratocytes is to use the tomographic configuration of the transmission microscope. Previously, it has been shown that by acquiring the two phase-shifted images, one can partially reject the out-of-focus light and recover the tomographic view in thick transparent ex vivo samples[34,50,51]. Here, we implemented a setup with two cameras having different levels of defocus in order to simultaneously capture the two optical phases. After subtracting the phase frames and enhancing the weak signal using NN[52], the resulting tomographic image successfully revealed the 15 μm stromal keratocyte nuclei (see Fig. 7g and Supplementary Movie 5).

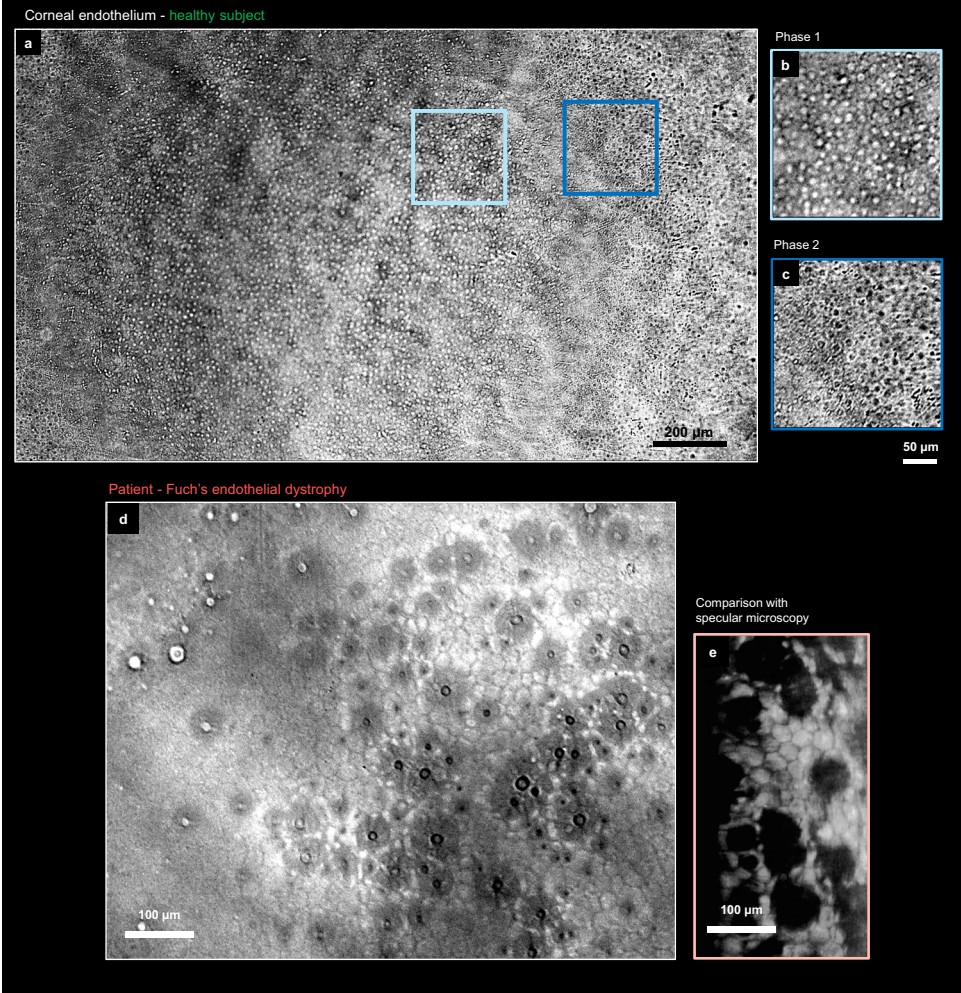

**Fig. 6 | Endothelium corneal layer in health and disease. a** Endothelial cells of healthy subject were visible across the extended 2 mm FOV achieved using 0.3 NA objective and ASI585 camera. **b**, **c** Transmission revealed the cell bodies with dark or bright interference contrast depending on their location relatively to the focal plane. **d** Endothelium of Fuch's endothelial dystrophy patient with guttata, captured with 0.4 NA objective and ASI432 camera. **e** Comparison with SM from the same subject, but a different (paracentral) corneal region. The central SM image was saturated due to substantial corneal haze. All presented images above are single camera frames without averaging.

Finally, the interference contrast enabled one of the most detailed looks into the microstructure of the crystalline lens in vivo (Fig. 8). The lens epithelium was composed of about 10–15 μm cells, which was in agreement with specular photography findings, reported using an experimental device in 1980s[53]. The cells were packed with a density of about 5000 cells/mm² (see the measurement details in Methods). The lens fibers were regularly packed and merged into the Y-suture, previously observed in vivo only with low-resolution OCT[54].

## Discussion

Fundamentally, this work demonstrates an interesting example of controlling the spatial coherence of light that is being scattered from random biological media, such as the sclera, for the benefit of imaging. Additionally, it highlights that the concept of the interference contrast in transmission extends beyond thin specimens and is applicable to thick tissues within the eye.

One of the instrument's primary applications is in surgical support. Today, the endothelial cell density measured by SM serves as an established clinical biomarker for the pre-screening of candidates for cataract and refractive surgeries. The transmission interference microscope has the potential to enhance the precision of this biomarker by bringing an extended FOV over which many more cells could be counted. Moreover, the extended FOV ensures that regions affected by the excessive guttae are not overlooked, enabling the identification and exclusion of patients with FED (300 million people affected[55]) from certain surgeries. Beyond the inclusion/exclusion criteria, the microscope could assist doctors in selecting the type of surgery best suited for the patient. For example, PRK surgery might be recommended over LASIK when the patient's cornea exhibits early symptoms of dystrophy, such as microfolds (striae) in the anterior cornea. Additionally, the high-resolution microscope can validate the next generation of refractive surgery instruments that rely on the microstructuring of the cornea[56].

Corneal infections are another major target for the clinical trials of the transmission microscope. This technology could eliminate the need for invasive corneal scrapping and contact CM[57], by offering a non-contact diagnostic approach that minimizes the risk of cross-contamination and is suitable for individuals with fragile corneas or children. Additionally, given the relatively low cost of its instrumentation - where the most expensive components are a consumer-grade USB camera (under $1000) and a microscope objective (under $3000) - and the robustness of its common-path interferometric design, this device presents opportunities for screening the population in the rural areas of developing countries, where infections are most prevalent.

A promising application for the transmission microscope lies in the assessment of corneal nerves, as there is currently no imaging

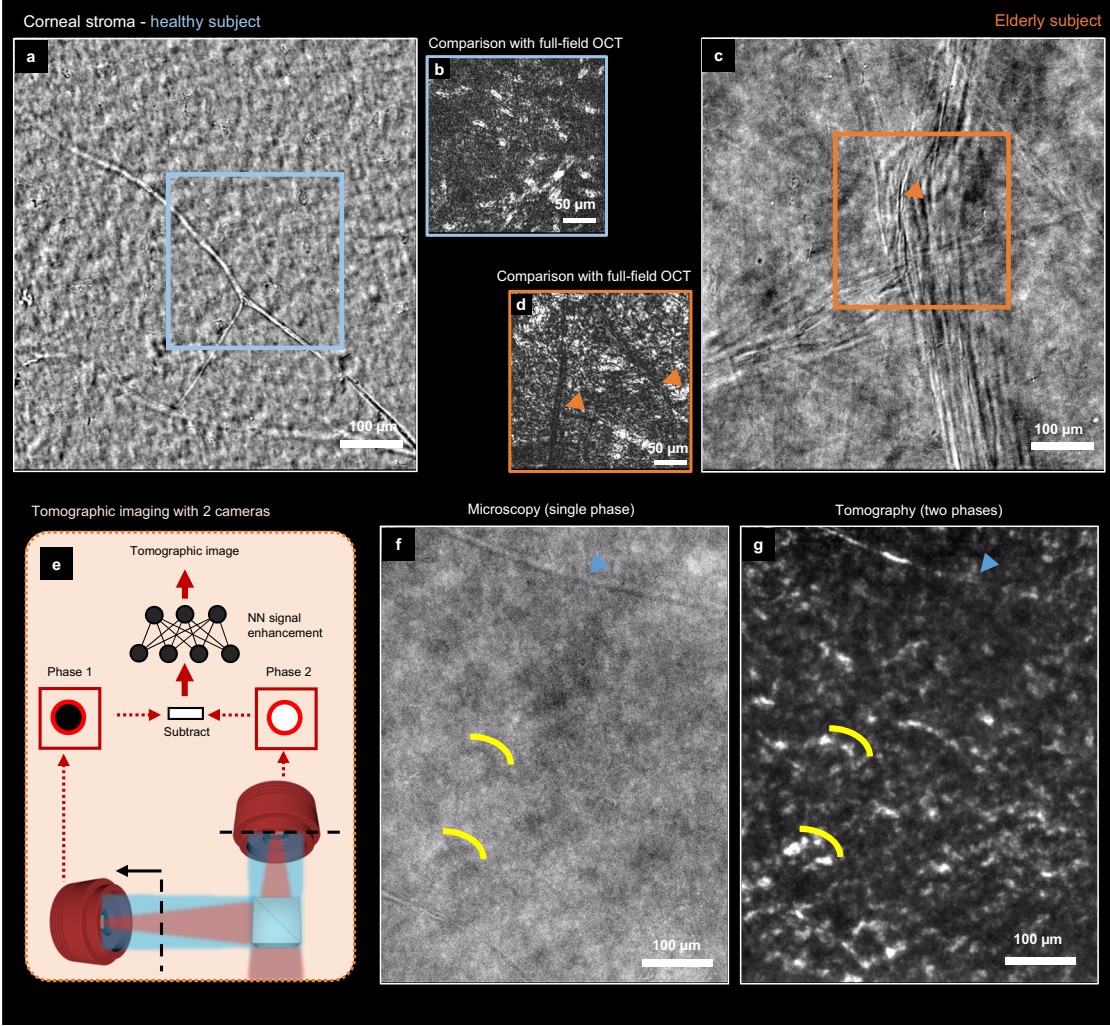

**Fig. 7 | Corneal stroma in transmission microscopy and tomography.**
**a** Transmission microscopy view of the healthy corneal stroma with visible stromal nerve. **b** Comparative reflection image from TD-FF-OCT is more contrasted and clearly shows nuclei of keratocytes. **c** Stroma of elderly subject had visible stripes, hypothesized to be corneal microfolds (striae). The images **a-c** were produced by averaging 15 frames. **d** The comparative reflection TD-FF-OCT image from the same subject. Striae are highlighted using red arrows. **e** Principle of tomographic imaging using 2 cameras. **f**, **g**, Comparison between transmission interference microscopy (left) and tomography (right). Yellow lines indicate the groups of keratocyte nuclei. Blue arrow points at stromal nerve. All images were captured using a 0.3 NA objective and an ASI432 camera. The images were cropped to facilitate comparison.

modality that allows their efficient, reproducible and non-invasive analysis. This would be particularly relevant in the pharmaceutical industry, where it can provide quantitative feedback on the efficacy of emerging treatments. For example, the effect of nerve growth factor (NGF) medication against the dry eye disease, which affects 20% of the world's population[58], could be assessed by measuring nerve density before and after treatment[59]. Additionally, nerve studies related to diabetes and brain-related diseases, which were previously conducted with CM[9,11], could be further facilitated with the transmission microscope, capable of capturing an extended FOV in a single short exposure and non-invasively. Moreover, the view of the crystalline lens offered by this technology could support the validation of breakthrough stem cell therapies developed for cataract surgeries[60]. Another compelling research direction is to examine the evolution of transmission images over time. In particular, advancing the work of CM on monitoring the activity of dendritic cells[61], could provide deeper insights into the vicious cycle of inflammation in dry eye disease[62].

Fundamental studies of light scattering in transmission microscopy are encouraged to better understand the observed image features. For example, such studies could help explain why Fuchs'

endothelial dystrophy and/or cataract enhance the visibility of endothelial cell edges, which are not apparent in healthy subjects.

In terms of current limitations, the method provides reduced contrast compared to CM. Due to the lack of confocality, it also has limited capability to reject out-of-focus light, which occasionally causes debris from the tear film to appear as defocused shadows in the intra-corneal sections. The unique contrast mechanism of our method —where the same cells may appear dark, bright, or even invisible—will require additional training for clinical specialists, as well as the development of adapted cell-counting algorithms. Alternatively, using two-phase reconstruction instead of single-phase microscopy offers another pathway to remove contrast ambiguity. Finally, incorporating a fixation target into the transmission microscope will be essential for accurate localization of the desired corneal region and for ensuring reproducibility in clinical studies.

The rapid evolution of NIR CMOS technology, propelled by the growing demand for security applications, is anticipated to bring a further increase in sensitivity and number of pixels in the coming years. These advancements could expand the FOV of microscopy to a degree approaching that of macro instruments in clinics, such as OCT and slit-lamp.

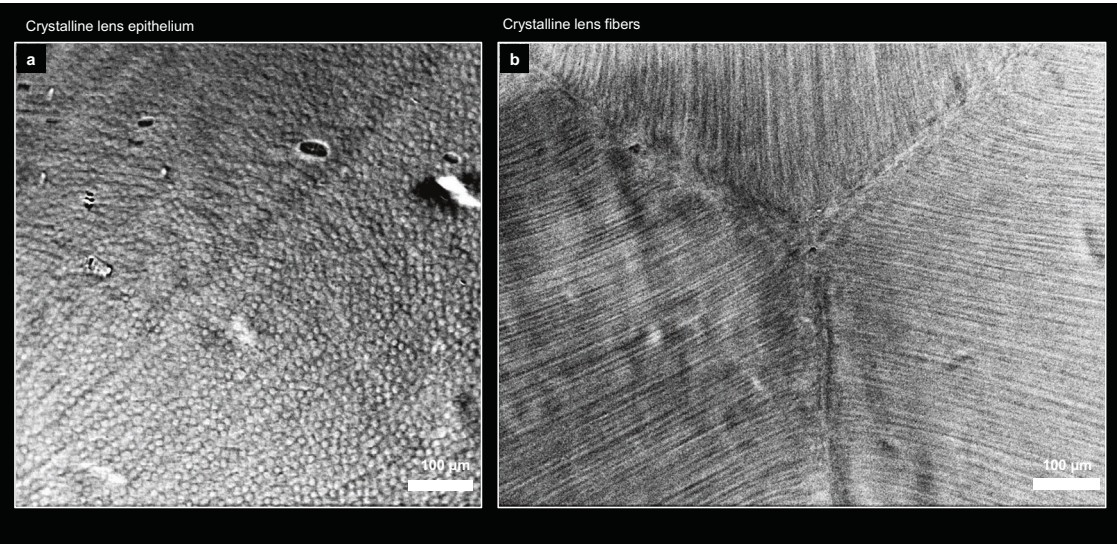

**Fig. 8 | Crystalline lens of the healthy human subject. a** Epithelium layer. **b** Fiber layer with visible sutures. The images were produced using a 0.3 NA objective and by averaging approximately 40 raw frames captured with the ASI432 camera.

## Methods

### Components of the prototype

The light from the NIR 850 nm LED chip (M850L3, Thorlabs, USA) was refocused onto the back-focal plane of the microscope objective with 1:1 magnification using a lens pair (LA1422-B, Thorlabs, USA). We used two interchangeable microscope objectives. For medium resolution, we employed the air objective (LMPLN10XIR, Olympus, Japan) with 10× magnification, 0.3 NA, 18 mm focal length (FL), and 18 mm working distance (WD). For high-resolution imaging, the air objective (plan Apo NIR, Mitutoyo, Japan), having 20× magnification, 0.4 NA, 10 mm FL, and 20 mm WD was utilized. The objectives converted the focused image of the LED at the back-focal plane into the quasi-collimated beam incident on the eye. The cornea and crystalline lens then focused the beam onto the retina and sclera at the back of the eye.

The size of the illumination spot on the retina/sclera was computed from the size of the LED chip and the focal lengths of the lenses, microscope objective, and the eye. Although the LED chip was 1 mm in size, the actual visible size of the LED was 1.8 mm due to the presence of the aspheric dome overlying the lens, which was confirmed using the ZEMAX ray data provided with the LED. Considering the common 17 mm value for the focal length of the eye[63], the spot size was 1.8 mm × 40 mm/40 mm × 17 mm/18 mm = 1.7 mm for an objective with 18 mm FL and 1.8 mm × 40 mm/40 mm × 17 mm/10 mm = 3 mm for an objective with 10 mm FL. Here, 40 mm refers to the focal length of the lenses used in a 1:1 magnification lens pair.

The light, scattered from the sclera, illuminated the anterior eye structures from the back, and the transmitted light was collected by the microscope objective and focused by the lens doublet onto the camera. We tested two budget astrophotography cameras, both with high quantum efficiency (QE) > 30% in NIR: ASI432 (ZWO, China), equipped with an imx432 sensor (Sony, Japan), or ASI585 (ZWO, China), equipped with an imx585 sensor (Sony, Japan). Using a camera with a high quantum efficiency was important because of the inherently weak light power originating from the sclera. Indeed, the sclera scatters light uniformly in all directions[64], meaning that only a small fraction of the scattered light can contribute to the back-illumination of the central cornea and eventually reach the detector. The ASI432 featured a global shutter with 1608 × 1104 pixels and a pixel size of 9 microns. It had an increased imaging speed of 120 frames/s (fps), which was beneficial for suppressing artefacts related to eye movements, including those caused by heartbeat, breathing, and fixational eye motions. The ASI585MC, on the other hand, had a rolling shutter

sensor and a slower 45 fps rate, but it featured 3840 × 2160 pixels (pixel size 2.9 μm), allowing for an extended FOV. Prior to the usage protected window was removed from the camera to increase transmittance in NIR.

The focal length of the tube lens was adapted to the pre-selected objective and pixel size of the camera to achieve sampling that adhered to the Nyquist criterion. Specifically, for a 0.3 NA objective and ASI585, we used a 100 mm doublet (AC254-100-B, Thorlabs, USA) that oversampled the resolution.

To collect the transmitted light and avoid the strong specular reflection from the corneal surface, the polarizing beam splitter (PBS252, Thorlabs, USA) was used. An additionally implemented linear polarizer (LPNIRE100-B, Thorlabs, USA) was necessary to produce a cleaner polarization state in the illumination. The optical setup was mounted on a clinical joystick stage capable of moving in X, Y, and Z dimensions (purchased from Lumedica, USA). The optical setup could also include a focus-tuneable lens (EL-16-40-TC-NIR-5D-C, Optotune, Switzerland) implemented in the detection path between the beam splitter and the tube lens. By synchronizing the tunable lens with the acquisition, the fly-through volume stack of the anterior cornea could be acquired in less than 1 s (Supplementary Movie 3). For this, we also cropped the sensor area by half, increasing the frame rate of ASI432 camera to 178 fps.

The full device was controlled by an Arduino Uno board (Arduino.cc, Italy) and a mini-PC (BXNUC9i9QNX, Intel, USA). The Arduino managed the pulsating of the LED with 0.5 ms time precision. The mini-PC, with a custom-written Python code, collected 8-bit or 10-bit images from the cameras via the rapid USB 3.1 port, processed them, and displayed the result in real-time. To address the limitations of the budget cameras lacking a trigger feature, we implemented a software-based rolling buffer. Only the image with the highest intensity within the rolling buffer, captured in the moment when the LED pulsation matched the camera exposure, was displayed on the screen.

### Simulation of wave propagation

We deployed the scalar wave equation simulator from ref. 65. Refractive indexes of the medium and lenses were 1 (corresponding to air) and 1.5 (corresponding to glass), respectively. The scatterer had a size comparable to the wavelength and an elevated refractive index of 2.5 for more contrasted visualization. The 4F system contained identical lenses with the scatterer and the camera being close to the focal plane of the respective lenses. In order to reduce the coherent artifacts in the

simulation, we generated the quasi-incoherent incident illumination by using a random assembly of particles ($n = 1.5$, diameter about ¼ of the wavelength) that randomized the phase of the incoming wavefront in time. The Fig. 1f, g, i were time-averaged. By positioning the assembly closer or further from the scatterer, one could also control the divergence of the illumination angles to which the scatter was exposed.

## Experiment with variable illumination size

We used an artificial eye (Model Eye for OCT, Modell-Augen Manufaktur, Germany) having a modular structure. The anterior part of the eye was interchangeable and configured with either a 4 mm or 8 mm pupil and a transparent or light-scattering lens, the latter resembling a cataract condition. The sclera below the retina was made of white, solid scattering material that was translucent to light.

In the experiment with variable illumination size, the 0.3 NA objective and ASI585 camera were used. The magnification of the light spot at the sclera was adjusted by varying the focal lengths of the lenses or by using an aperture in the illumination arm. Sub-figures in Fig. 3b were acquired using the lens pairs of 8 mm/50 mm, 16 mm/50 mm, 30 mm/50 mm, and 50 mm/50 mm, respectively. In case of the smallest illumination size, we used 50 mm/50 mm lenses with an additional 200 μm aperture in front of the LED. The final size of the magnification spot was calculated using the formulas discussed above, with one modification: the artificial eye had a longer focal length of 25 mm due to a smaller corneal curvature. In the example of the 200 μm aperture the projected light spot was 0.2 × 25 mm/18 mm ≈ 0.3 mm, while in the case of 8 mm/50 mm it was 1.8 mm × 50 mm/8 mm × 25 mm/18 mm ≈ 15 mm. We confirmed the illumination sizes experimentally by replacing the posterior part of the eye with the second camera (ASI432).

The objective was focused on a single corneal plane of the artificial eye. In order to reconstruct the axial cross-section image and 3D volume, the artificial eye was moved axially using a motorized stage (X-DMQ12P-DE52, Zaber, Canada). To facilitate comparison, the axial intensity profiles were normalized between 0 and 1, based on the maximal and minimal intensity values measured in the most contrasted case of the 0.3 mm spot size. The relation between the contrast and illumination spot in Fig. 3d was obtained by calculating the difference between the maximal and minimal values of the axial profile for each spot size.

## Light diffusion in the sclera

The secondary illumination source re-emitted from the sclera is larger than the spot size of the incident illumination due to light diffusion in the sclera. From the reduced scattering coefficient of 30 cm$^{-1}$ for the sclera[38,39], the transport mean free path was computed as $l^* = 1/30 \, cm^{-1} = 0.33 \, mm$. Considering forward and backward propagation through the 1 mm thick sclera the increased spot size is given by: $D = 2 \cdot \sigma_{total} = 2 \cdot \sqrt{\sigma_0^2 + 2 \cdot z \cdot l^*}$, where $\sigma_{total}$ is the total variance, $\sigma_0$ is the initial spot radius and $z$ is 1 mm sclera thickness.

## Imaging of human subjects

The instrument was validated by imaging two male (32 y/o, 28 y/o) and two female (41 and 43 y/o) healthy subjects, as well as one male patient (78 y/o) and one elderly subject (82 y/o) with corneal abnormalities. Prior to the research exam, all subjects were screened using the conventional slit-lamp in the hospital, which confirmed their corneal conditions. In particular, the patient displayed advanced Fuch's endothelial dystrophy and cataract. The elderly subject exhibited mild corneal haze typical of this age group. In addition to the exam with the transmission microscope, subjects were selectively imaged with a clinical specular microscope (CEP-530, Nidek, Japan), a time-domain full-field OCT research instrument for clinical research[13], or/and with a confocal microscope (HRT3 RCM, Heidelberg engineering, Germany). These high-resolution devices, which operate in reflection, served as a comparison reference for the transmission images.

The subjects were asked to place their head on a clinical chinrest with forehead support. The examination was performed in a dark room to allow natural dilation of the pupil. No medications were introduced in the eye. In accordance with the French regulators, the approval for the study was obtained from Comité de Protection des Personnes (CPP) Sud-Ouest et Outre Mer III, study number 2024-A01040-47. Each subject has provided informed consent for participation in the study. Imaging was non-contact, with a comfortable 2 cm distance between the objective and the eye. The device was mounted on a clinical joystick stage, which enabled alignment with the patient's eye along three axes. The instrument was emitting pulsed NIR illumination with radiant exposure adhering to the international light safety standard ISO 15004-2:2007[66]. The light power was distributed across both time and space. More precisely, the 10 ms light pulse was followed by the 90 ms interval without light. At the retinal plane, the illumination was spread across the full-field area of 1.7 mm that was considerably larger than the micrometric focused spot, typical for scanning OCT devices. As a result, the total incident power of 100 mW (per pulse) produced a time-averaged weighted retinal irradiance of 100 mW × 0.01 s/(0.01 s + 0.09 s)/(1.7 cm)$^2$ × R ≈ 220 mW/cm$^2$, where R is the spectral weighting coefficient (R = 0.63 for 800 nm from Table A1, ISO 15004-2[66]). The irradiance value was below the maximum permissible limit of 1.2/1.7 W/cm$^2$ = 700 mW/cm$^2$ (Table 4, ISO 15004-2[66]). Similarly, the safety on the short time scale (Table 6, ISO 15004-2[66]) was confirmed. For t = 20 s the weighted radiant exposure was 100 mW/(1.7 mm)$^2$ × 20 s × 0.01 s/(0.01 s + 0.09 s) × R ≈ 4.4 J/cm$^2$, smaller than the permissible limit of 10/1.7 × (20 s)$^{3/4}$ × (N$_{pulses}$)$^{-1/4}$ J/cm$^2$ ≈ 14.7 J/cm$^2$. Here the N$_{pulses}$ is the number of pulses during 20 seconds. The exposure of the anterior eye was relatively weaker due to the collimation of the beam incident on the eye. Indeed, the measured irradiance of 5 mW/mm$^2$ at the corneal plane resulted in time-averaged irradiance of 50 mW/cm$^2$, below the 100 mW/cm$^2$ limit (Table 4, ISO 15004-2[66]). On the short time scale of 20 s, the exposure of 1 J/cm$^2$ was smaller than the maximal permitted 3.8 J/cm$^2$ (Table 6, ISO 15004-2[66]). We also verified the corneal and retinal safety for all the times below 20 s. For the extended light exposure calculations see Supplementary Information file.

In an alternative mode, we illuminated the eye with a 1 s exposure followed by a 19 s interval without light. This mode was useful for acquiring more frames with minimal time delay, thus allowing for improved the SNR through image averaging. Repeating the calculations above with t = 1 s (the highest exposure condition) confirms that the short-time-scale radiant exposures for retina of 2.2 J/cm$^2$ and cornea of 0.5 J/cm$^2$ were below their respective limits of 5.8 J/cm$^2$ and 1.8 J/cm$^2$. The same holds true for the time-averaged exposure, where the retinal 170 mW/cm$^2$ and corneal 25 mW/cm$^2$ values are below their respective limits of 700 mW/cm$^2$ and 100 mW/cm$^2$.

Refractive errors and the age of the subject can have a minor impact on the safety calculations. For instance, refractive errors tend to spread the light over a larger retinal area, which can improve the safety margin of the device. Conversely, in young children (under 10 years old), the safety margin may be slightly reduced due to differences in ocular geometry. Specifically, young children often have a shorter effective focal length (around 15 mm) and a correspondingly shorter axial eye length. As shown in the Supplementary Information file, this leads to a smaller illumination spot on the retina—1.5 mm instead of 1.7 mm. In the most limiting case of a 1-second exposure followed by a 19-second break, this results in a short-time radiant exposure of 2.8 J/cm$^2$, which is 2.4× below the safety limit of 6.6 J/cm$^2$—a slight reduction compared to the 2.6× margin observed in a normal adult eye.

## Image processing

The processing steps were dependent on the camera type. The ASI585 was a color camera, which required a Gaussian blur with minimal kernel size (3 × 3 pixels) in order to uniformize the illumination mismatch between the different segments of the Bayer filter. Conversely, the ASI432 was a mono camera with a front-illuminated architecture, which introduced fixed pattern noise. This noise was removed by subtracting the average from the stack of moving images of the eye. Additionally, for both cameras, we applied high-pass filter with a large kernel (121 × 121 pixels) that suppressed the bright uniform background. In selected cases, for example, in crystalline lens data, the images were averaged in order to increase SNR.

## Quantitative image analysis

Characteristic cell sizes in the images were quantified by manually measuring diameters of several cells in Fiji[67] and calculating the mean. The density of cells was calculated by counting the cells using the Point tool of Fiji[67] and dividing their number by the area in which they were observed.

## Neural network architecture

We employed the U-net neural network architecture, as detailed in ref. [52], for both segmentation and signal enhancement tasks. The segmentation network was trained using the in vivo transmission data from the sub-basal nerve layer. Ground truth images were generated by segmenting the nerves in a semi-automated way using the NeuronJ plugins of Fiji[67]. The signal enhancement neural network, trained on full-field OCT data, was deployed from ref. [52] and directly applied without retraining to enhance the weak tomographic transmission signals.

## Statistics and reproducibility

In vivo human data were acquired over multiple imaging sessions, with several image sequences obtained from each subject. Although the exact same corneal location was not always imaged—due to the lack of a fixation target—the same types of corneal features were reproducibly observed in the same subjects across different days.

## Reporting summary

Further information on research design is available in the Nature Portfolio Reporting Summary linked to this article.

## Data availability

All images that support the findings of this study are presented in the manuscript.

## Code availability

The neural network code used in this article is available in GitHub repository: https://github.com/vmazlin/i2i.

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

## Acknowledgements

We thank Daniel Royer for fruitful discussions. We also would like to acknowledge the funding support of Agence Nationale de la Recherche grants: ANR-22-CE19-0018 (V.M., C.B., and Mathias Fink), ANR-22-CE45-0005 (V.M., C.B.), and ANR-17-CONV-0005 (Q-life project TIOS, obtained by V.M.).

## Author contributions

Idea and concept (V.M., S.A., A.C.B.), physical theory and simulations (V.M., S.A., A.C.B.), development of the device and experiments (V.M., S.A., A.C.B.), software for acquisition and image analysis (S.A., V.M.), clinical guidance (W.G., C.B.), interpretation of biomarkers (W.G., C.B., V.M., S.A.), neural network processing (V.M., S.A.), safety analysis (V.M.), prototype 3D design and printing (S.A., V.M.). V.M. has written the manuscript, S.A. prepared the supplementary materials, and all authors contributed with their feedback.

## Competing interests

V.M., S.A., A.C.B.: Patent; S.A.: Part-time salary from the company (SharpEye); V.M., A.C.B.: Ownership of the company shares (SharpEye). The remaining authors declare no competing interests.
