## [Transparent Peer Review file · Nature Communications]

Transmission interference microscopy of anterior human eye

Corresponding Author: Dr Viacheslav Mazlin

Version 0:

Reviewer comments:

Reviewer #1

(Remarks to the Author)

In this manuscript, Alhaddad et al. demonstrate the transmission interference microscopy of the anterior segment of the human eye. Such an imaging method has already been shown by the Authors (Biomedical Optics Express 13(8), 4190 (2022)). Their previous work demonstrated full-field optical transmission tomography for imaging single-cell diatoms and ex vivo biological samples, including porcine and macaque corneas. The novelty of this manuscript lies in the application for in vivo imaging of the human cornea. As transmission geometry is challenging in ocular imaging, the imaging beam is focused on the back of the eye, and back-scattered light is used for transmission interference microscopy of the cornea. They also show that the size of the illuminated retina/sclera (i.e., the size of the secondary light source) can work similarly to the aperture on the condenser lens in conventional microscopy, improving contrast.

The proposed imaging modality is demonstrated in healthy subjects and two clinical patients with anterior eye abnormalities. The Authors also demonstrate the clinical prototype of the system.

While in general, the study's data support the conclusion, there are several points that require further discussion.

1. While the instruments allow one to image corneal sub-basal nerves and nerve density was in agreement with confocal microscopy, the comparison is not convincing, as images in Fig. e,f demonstrate different portions of the cornea.
2. Phase 1 and Phase 2 are both present in the images, and endothelial cells appear as bright and dark spots. It seems that there is a region 'in-between' where the contrast might be lost or is significantly reduced. This is visible in Fig. 4a. Please discuss this point, as this may lead to some features that might not be visible in the image.
3. In FED patients the comparisons between transmission interference microscopy, FF-OCT, and SM are not entirely convincing, as it was done on different subjects. Indeed, transmission interference microscopy provides the clearest insight into the imaged structures, which is nice. However in provided images, we are looking at different structures, so it is difficult to draw robust conclusions.
4. Why the light source is pulsed? Is 100 mW the average power (as measured by the power meter) or the peak power of the pulse?

Methodology

The results are reproducible, the manuscript contains sufficient amount of details. The methodology is sound, however, the comparison with other methods is not convincing as different subject and different regions were compared.

Significance

The discussion in the paper highlights the significance of the results. The most important advantages are non-contact diagnostics larger FOV (compared to CM), and simpler setup (compared to FF-OCT). Many applications could benefit from such a technology, including surgical support and non-contact diagnostics of anterior eye abnormalities.

Suggested improvements

1. "In vivo cells and nerves are revealed with increased contrast across all corneal layers and within an extended 2 mm field of view, 7× larger than the clinical state-of-the-art." It is unclear what is the clinical state-of-the-art in this context. Later on, the Authors mention that their instrument has a 10x higher viewing area than a confocal microscope. In another place, they mention that their instrument has a 16× larger viewing area than the state-of-the-art SM.
2. Supplementary Fig. 1 b,c shows the comparison of the imaging of USAF targets in reflection and transmission. Some additional information is necessary to properly understand this figure, i.e., what kind of USAF target is used? What is the power incident to the sample in each configuration? How much power is back-scattered by the scattering paper?

Reviewer #2

(Remarks to the Author)

The authors have developed and tested an affordable microscope for corneal imaging. A key achievement is the modification of the method described in "In vivo corneal and lenticular microscopy with asymmetric fundus retroillumination" by Weber and Mertz, allowing them to obtain high-resolution images with a significantly larger field of view. This innovation could enable better screening in resource-limited settings.

The work has substantial significance, particularly for ophthalmology and global health applications. By providing a cost-effective solution for corneal imaging, this device has the potential to improve accessibility to eye care in rural and developing areas where corneal infections are more prevalent. Compared to established literature, the device offers a larger field of view, which could be a valuable advantage over existing commercial systems.

While the work demonstrates promising results, some claims—such as the significantly larger viewing area and cost-effectiveness—would benefit from additional references or quantitative comparisons to commercially available systems.

Furthermore, some numerical discrepancies in calculations (e.g., NA values, spot size estimations) require clarification. Additional safety considerations for subject's head alignment (before imaging) and more detailed explanations of equations used in safety assessments would help support the conclusions.

There are no fundamental flaws that would prohibit publication, but several areas require revision:

- Some calculations do not align with expected values and need clarification.
- The methodology for measuring endothelial cell density and nerve density using a neural network requires justification over more traditional segmentation approaches.
- Claims regarding resolution and contrast control require additional experimental validation, such as measuring resolution with a resolution target.

Overall, the methodology appears sound, with a well-thought-out approach to modifying existing techniques.

The manuscript lacks some critical details necessary for full reproducibility. Specifically:

- Several key parameters used in safety calculations are not fully described, making it difficult for readers to verify the approach without direct access to ISO 15004-2:2007 and ANSI Z80.36-2021 standards.
- Additional methodological details, such as how cell diameters were measured and how image acquisition settings impact resolution, would improve transparency.

The manuscript presents an important and promising advancement in affordable corneal imaging. With revisions to improve clarity, address numerical inconsistencies, and provide additional methodological details, the work has the potential to make a significant contribution to the field.

Below, I provide detailed feedback and suggestions for improving the clarity and rigor of your manuscript. I hope these comments will help refine your work and enhance its presentation:

1. Lines 85-86: The sentence appears unclear. I recommend rewriting it for better clarity.
2. Line 68: Please replace "focus at" with "focus on" for correct phrasing.
3. Lines 85-101: The terms "first lens" and "second lens" in the description of the setup are vague. Consider assigning specific names to the optical elements and using those consistently when describing the setup and experiments.
4. Lines 98-99: In the sentence "The sign of the contrast, denoting constructive or destructive interference, can be alternated by moving the camera, the scatterer, or the lenses," please specify the direction and axis of movement.
5. Line 138: The refractive index of the cornea-lens ($n_{\text{cornea-lens}}$) is missing.
6. Line 140: The refractive index in the anterior eye (n) is also not provided.
7. Lines 141-143: How were the NA values calculated? Assuming the only variable is the source radius, the ratio of the two NA values should equal $15/0.3 = 50$, but in your manuscript, it equals 40. Could you clarify this discrepancy?
8. Line 148: The phrase "light of higher coherence" is ambiguous. Do you mean spatial coherence? The same applies to Line 282: "controlling the coherence of light." Please specify the type of coherence.
9. Lines 155-156: The enlarged spot size of 2.4 mm (reported in Methods) does not match the calculated value based on the equation in Line 400: $\sqrt{(1.7^2 + 2 \times 1 \times 0.33)} = 1.88$ mm. Similarly, the value of 3.4 mm in Line 160 could not be reproduced (I calculated 3.1 mm). Could you explain these differences?
10. Lines 184-188: This section would benefit from a summary of the results presented in the supplementary materials to tie them back to the main text.
11. Line 203: How were cell diameters measured? I could not find this information in the Methods section.
12. Lines 207-208: Why did you use a U-net neural network for nerve density evaluation rather than a more classical segmentation approach?
13. Lines 211-212: The claim of achieving a "10× larger viewing area" needs a reference. Additionally, the HRT3 device by Heidelberg Engineering reportedly has a field of view of $400 \times 400 \mu\text{m}$, which is not 10 times smaller than the reported 1 mm field of view (source: <https://business-lounge.heidelbergengineering.com/ie/en/products/hrt3-rcm/hrt3-rcm/publications/>).
14. Lines 215-216: How was the endothelial cell density measured?
15. Lines 225-226: The statement "16× larger viewing area than the state-of-the-art SM" lacks a reference.
16. Figures 4 and 5: The green text is difficult to read. I suggest changing the font color for better visibility.
17. Figure 4e: Why are images from a different subject shown? It would be preferable to show images from the same subject and location. If this is not possible, please provide detailed information about the "other subject" and the setup used.
18. Lines 275-276: How was the cell density measured?
19. Lines 299-301: Could you provide an estimated cost of the device and specify how it compares to commercially available systems?
20. Line 333: The eye focal length value requires a reference and the name of the eye model used.
21. Lines 334-335: For a 10 mm focal length objective, the calculation appears incorrect. Should it be $1.8 \text{ mm} \times 17 \text{ mm} / 10 \text{ mm} = 3.06 \text{ mm}$?
22. Lines 338-339: It would be helpful for readers to include information about the sensor pixel size in the manuscript.
23. Line 339: Replace "Using a camera with high QE ..." with "Using a camera with a high quantum efficiency (QE) ..." for

clarity.

24. Lines 343-344: What kind of eye movements are you referring to? Please elaborate.

25. Line 351: If you are providing manufacturer details for other components, the corresponding information here is missing.

26. Line 367: Where does the value of the lens refractive index originate?

27. Lines 375-378: According to the manufacturer's website, the light-scattering lens imitates a patient's natural lens, not a cataract lens (source: <https://modell-augen-manufaktur.de/wp-content/uploads/2023/06/brochure-OCT-model-eye.pdf>). Did you use a different, more scattering lens? Please clarify.

28. Lines 420-434: The safety considerations outlined in this section rely on ISO 15004-2:2007 and ANSI Z80.36-2021 standards, which may not be readily accessible to all readers. To improve clarity and reproducibility, I suggest elaborating further on the equations and values used in your analysis. Providing additional details would help readers follow and verify your approach without requiring direct access to these standards.

I hope these comments are helpful and encourage further refinement of your manuscript. Please feel free to reach out if any points require additional clarification.

Best regards,
Reviewer

Reviewer #3

(Remarks to the Author)

Imaging the anterior segment of the eye with cellular-level details remains a technological challenge. In this paper, the authors proposed a transmission interference microscopy technique in which the backscattered light from the sclera serves as a secondary source of illumination and imaging of the anterior eye. The image of individual layers of the eye within the focal plane of the microscopic objective was formed by the interference between the backscattered light from cellular features and the transmitted light non-affected by the cellular features. In this work, the authors also evaluated the impact of the size of the secondary source (e.g., the size of the illumination spot on the sclera) on the performance (contrast, resolution, DOF) of the imaging system. The proposed method enabled a 2-mm FOV of the anterior eye with cellular resolution, much wider than that of standard clinical and research systems. The system's performance was demonstrated through imaging of four young healthy subjects, one FED patient with cataract, and one elder subject. The paper is scientifically sounded and appropriate for publication in Nature Communication provided that authors address the concerns below:

1. The formulas in lines 138-140 indicate that the higher the NA_i , the higher the NA_{eff} , which seems to contradict the experimental observation (e.g., in lines 141-143). It is not clear why the increased contrast of the interference with the size of the illuminated sclera is the result of the NA_i , i.e., the numerical aperture of the illuminated sclera visible to the scatterer in the anterior eye. Considering the sclera as a condenser instead of a scattering element may be a strong assumption. I encourage the authors to explore other alternatives. It is likely that the improved contrast of the interference with the size of the illuminated sclera simply has to do with the spatial coherence of the secondary source instead of the NA_i the way it is defined by the authors. The larger the source, the lower the spatial coherence and the lower the interference contrast due to the scrambling of the phase from different points of the source. This consideration also affects the evaluation of the impact of pupil size on the contrast, as well as the explanation of different scattering behaviors in lines 175-183 (major).
2. Despite the improved performance with the reduced size of the illuminated sclera, care must be taken to prevent retinal damage. Operating the system with an irradiance that is halfway the MPE limit can be risky as several factors including the refractive error of the patient's eye can quickly put the patient at risk of retina phototoxicity (major).
3. Lines 218-220: The description of guttae characteristics must be supported by additional data. The external 30-100 μm dark halo could well be the result of shadow created by guttae in the sclera illuminated beam (major).
4. Since the imaging performance relies on the transmission of the eye, it will be important to evaluate the utility of the system in disease conditions such as cataracts as well as in highly aberrated eyes (minor).
5. Images have different sizes and it is not specified on what subject such or such images (e.g., figures 3, 6) were collected, nor whether images were from the same imaging session. Please provide images with similar sizes if possible for consistency. Otherwise, clarify the discrepancy in image size (minor).
6. Revise the beginning of the abstract to provide a context to the work (minor).
7. Line 78: Remove the word "curiously" as this effect has been previously reported by Weber and Mertz (minor).
8. Add a reference in line 63 (minor).

Version 1:

Reviewer comments:

Reviewer #1

(Remarks to the Author)

After carefully reviewing the revised manuscript, I find that all of my previous comments have been adequately addressed by the authors. Additionally, I have examined the responses to the comments raised by the other reviewers, and to the best of my knowledge, those concerns have been appropriately resolved as well. Overall, the manuscript has significantly improved. I have only one minor comment remaining, detailed below.

1. "Conversely, with a point source, the scatterer is illuminated by light of higher spatial coherence, originating from a single illumination angle." – please consider changing to "smaller angle".

Reviewer #2

(Remarks to the Author)

In the manuscript "Transmission interference microscopy of anterior human eye" by Alhaddad et al. present a novel application of transmission interference microscopy for in vivo imaging of the human cornea, building upon the authors' previous work on ex vivo biological samples. The innovative use of backscattered light from the retina/sclera as a secondary illumination source is both technically elegant and practically significant. The demonstration of the role of the secondary source size in contrast modulation adds further value to the methodology. Furthermore, the implementation of this system in a clinical prototype, tested on both healthy and pathological cases, provides convincing evidence of the potential clinical relevance.

The comparison with prior literature, especially the modification of the technique described by Weber and Mertz, and the achievement of a significantly wider field of view, underscores the originality of the work. The proposed device's affordability and portability also suggest important implications for global ophthalmic care, especially in underserved regions. Overall, the manuscript is scientifically sound, well-motivated, and clearly presented. The methods are described in sufficient detail to allow reproduction, and the conclusions are supported by the presented data.

I would like to thank the authors for their thoughtful and thorough responses to the reviewers' comments and for addressing the suggestions made in the previous round of review. In particular, I appreciate the detailed clarifications and calculations provided regarding safety considerations, which significantly enhance the credibility and completeness of the manuscript.

I only have three minor editorial suggestions before the manuscript can proceed to publication:

Line 180 – I believe the correct reference is "Fig. 4c" (not Fig. 3c).

Line 193 – It seems the authors meant "Fig. 3d" instead of "Fig. 2d).

Line 274 – Please check this reference; it likely should be "Fig. 6e" (not "Fig. 4e").

Once these minor corrections are addressed, I believe the manuscript is ready for publication.

Reviewer #3

(Remarks to the Author)

The authors convincingly addressed most of the reviewers' comments. The manuscript clarity has been substantially improved as a result of these changes and is deemed for publication. I only have a few minor comments:

- When comparing TIM with existing devices, especially when imaging the central cornea of the same participant, one would expect to identify the cellular features (e.g., corneal nerves in Fig.5 and guttae in Fig. 6) from the CM/SM image on the TIM image, given its larger FOV.
- It is still puzzling to me why endothelial cell boundaries are visible in the Fuchs' patient (Fig. 6d) but not in the normal participant (Fig. 6a). Further discussion would be helpful!

Dear Reviewers,

We would like to thank you for your thorough and insightful comments, as well as for your encouraging feedback on our method, which differs substantially from conventional reflection-based approaches. Your comments were invaluable for enhancing the clarity of this work.

Please find enclosed our point-by-point responses to your comments, along with the corresponding revisions made to the manuscript. Both the clean and red-lined versions of the revised manuscript are provided for your convenience.

Viacheslav Mazlin

on behalf of all co-authors

Reviewer 1:

In this manuscript, Alhaddad et al. demonstrate the transmission interference microscopy of the anterior segment of the human eye. Such an imaging method has already been shown by the Authors (Biomedical Optics Express 13(8), 4190 (2022)). Their previous work demonstrated full-field optical transmission tomography for imaging single-cell diatoms and ex vivo biological samples, including porcine and macaque corneas. The novelty of this manuscript lies in the application for in vivo imaging of the human cornea. As transmission geometry is challenging in ocular imaging, the imaging beam is focused on the back of the eye, and back-scattered light is used for transmission interference microscopy of the cornea. They also show that the size of the illuminated retina/sclera (i.e., the size of the secondary light source) can work similarly to the aperture on the condenser lens in conventional microscopy, improving contrast. The proposed imaging modality is demonstrated in healthy subjects and two clinical patients with anterior eye abnormalities. The Authors also demonstrate the clinical prototype of the system. While in general, the study's data support the conclusion, there are several points that require further discussion.

We thank the reviewer for comprehensively highlighting the results and the novelty of our work.

1. While the instruments allow one to image corneal sub-basal nerves and nerve density was in agreement with confocal microscopy, the comparison is not convincing, as images in Fig. e,f demonstrate different portions of the cornea.

Localizing the precise spot of interest is well-known to be complicated when using the confocal technique [1,2]. This difficulty arises from the very small field of view (only 0.25 mm) and the contact nature of the method, where additional pressure on the eye can cause the corneal layers to curve, further reducing the useful imaging field. Consequently, the results strongly depend on the clinical operator's expertise. The images in the manuscript represent the best that could be obtained by our orthoptist at the hospital.

[1] Niederer RL, McGhee CN. Clinical in vivo confocal microscopy of the human cornea in health and disease. Prog Retin Eye Res. 2010 Jan;29(1):30-58. doi: 10.1016/j.preteyeres.2009.11.001. Epub 2009 Nov 26. PMID: 19944182.

[2] Chidambaram, J.D., Prajna, N.V., Palepu, S. *et al.* Cellular morphological changes detected by laser scanning *in vivo* confocal microscopy associated with clinical outcome in fungal keratitis. *Sci Rep* 9, 8334 (2019). <https://doi.org/10.1038/s41598-019-44833-9>

We have added to the text:

“The measured density of 22 mm/mm² was in agreement with the state-of-the-art CM results from the same subject (different corneal location) (Fig. 5f)”

We have also added to the caption of Figure 5 (in the original manuscript it was Figure 3):

“Comparison with confocal microscopy image obtained on the same subject (different corneal location). Finding the same corneal location was notably difficult with CM due to its small field-of-view and contact nature, where additional pressure on the cornea can cause the layers to curve, further reducing the useful imaging field. Consequently, the results were strongly dependent on the clinical operator's expertise.”

2. Phase 1 and Phase 2 are both present in the images, and endothelial cells appear as bright and dark spots. It seems that there is a region ‘in-between’ where the contrast might be lost or is significantly reduced. This is visible in Fig. 4a. Please discuss this point, as this may lead to some features that might not be visible in the image.

We thank the reviewer for raising this important point.

We have added to the text:

“Importantly, in the exact optical focus, the phase difference between the transmitted and diffracted waves is minimal, rendering the cells invisible. To avoid ambiguity, cell density was calculated by counting only the bright cells and dividing their number by the area in which they were observed.”

We have also added a paragraph to discussion section about the limitations of the method:

“In terms of current limitations, the method provides reduced contrast compared to CM. Due to the lack of confocality, it also has limited capability to reject out-of-focus light, which occasionally causes debris from the tear film to appear as defocused shadows in the intra-corneal sections. Finally, the unique contrast mechanism of our method—where the same cells may appear dark, bright, or even invisible—will require additional training for clinical specialists, as well as the development of adapted cell-counting algorithms. Alternatively, using two-phase reconstruction instead of single-phase microscopy offers another pathway to remove contrast ambiguity.”

3. In FED patients the comparisons between transmission interference microscopy, FF-OCT, and SM are not entirely convincing, as it was done on different subjects. Indeed, transmission interference microscopy provides the clearest insight into the imaged structures, which is nice. However in provided images, we are looking at different structures, so it is difficult to draw robust conclusions.

Following the recommendation of the reviewer, we performed a specular microscopy (SM) exam on the same FED patient. However, due to corneal edema and the resulting increase in corneal haze, the SM image from the central corneal region was saturated, and endothelial cells could not be observed (Additional Fig. 1 in this file). To enable comparison with the transmission results, an SM image was instead captured from a paracentral region.

Additional Fig. 1. Specular microscopy images from different corneal regions of the same FED subject.

We have added this SM image to the article:

“... e, Comparison with specular microscopy from the same subject, but a different (paracentral) corneal region. The central SM image was saturated due to substantial corneal haze...”

Note:

The comparison dataset (transmission / specular microscopy) with dozens of subjects is currently being collected as part of the clinical trial at 15-20 Hospital in Paris and will form the basis of the first *clinical research article* on the new transmission method. This article will enable robust conclusions to be drawn.

On the other hand, those results would be out of scope (and beyond the maximum text length) for inclusion in the current manuscript, which is focused on the first presentation of the new eye imaging method that introduces a new contrast. In this context, the Transmission and SM images captured in the same subject but in different corneal regions serve well the purpose of highlighting the physical contrast differences between the traditional reflection methods and the new transmission technique.

4. Why the light source is pulsed? Is 100 mW the average power (as measured by the power meter) or the peak power of the pulse?

Pulsation of the light source is essential to stay within the maximal permissible exposure limits set by the ISO 15004-2 ocular light safety standard. The 100 mW power is the peak power of the pulse.

We have added clarification in the text:

“As a result, the total incident power of 100 mW (per pulse) produced a time-averaged weighted retinal irradiance of $100 \text{ mW} \times 0.01 \text{ s} / (0.01 \text{ s} + 0.09 \text{ s}) / (1.7 \text{ cm})^2 \times R \approx 220 \text{ mW/cm}^2$, where R is the spectral weighting coefficient (R = 0.63 for 800 nm from Table A1) according to ISO 15004-2:2007.”

Additionally, the detailed safety calculations can now be found in *Supplementary file*.

Methodology

The results are reproducible, the manuscript contains sufficient amount of details. The methodology is sound, however, the comparison with other methods is not convincing as different subject and different regions were compared.

Please see our extended answers to questions 1 and 3 above.

Significance

The discussion in the paper highlights the significance of the results. The most important advantages are non-contact diagnostics larger FOV (compared to CM), and simpler setup (compared to FF-OCT). Many applications could benefit from such a technology, including surgical support and non-contact diagnostics of anterior eye abnormalities.

We thank the reviewer for pointing to the significance of the obtained results.

Suggested improvements

1. “In vivo cells and nerves are revealed with increased contrast across all corneal layers and within an extended 2 mm field of view, 7× larger than the clinical state-of-the-art.” It is unclear what is the clinical state-of-the-art in this context. Later on, the Authors mention that their instrument has a 10x higher viewing area than a confocal microscope. In another place, they mention that their instrument has a 16× larger viewing area than the state-of-the-art SM.

We agree with the reviewer that our usage of several metrics and benchmarks complicated the message. Specifically, the confusion arose from using different performance metrics (FOV versus imaging area) and from making comparisons against different state-of-the-art benchmarks (Clinical versus Research).

We unified the comparison to be with the clinical state-of-the-art and in terms of FOV:

“Moreover, new transmission design gives access to 2 mm x 1 mm FOV, which is about 5× larger than the SM (0.25 mm × 0.55 mm)⁴⁷ and CM (0.4 mm × 0.4 mm)⁴⁸ in clinics.”

2. Supplementary Fig. 1 b,c shows the comparison of the imaging of USAF targets in reflection and transmission. Some additional information is necessary to properly understand this figure, i.e., what kind of USAF target is used? What is the power incident to the sample in each configuration? How much power is back-scattered by the scattering paper?

A 1951 USAF target (R1DS1P1, Thorlabs, USA) was used as a sample. Both transmission and reflection tests were conducted using the same low optical power (a few mW). The scattering paper, composed of cellulose, scatters almost all illumination (90%) in NIR [1].

[1] https://www.sshade.eu/data/SPECTRUM_OP_20181101_06

We have moved this supplementary figure to the main text (new Fig. 2) and added the clarifying information about the details and the purpose of that experiment

The photo of the prototype and its performance with a target are shown in Fig. 2. When the target was placed at the objective focal plane, minimal light was detected. This is because specular reflections from the target were filtered out by a polarization beam splitter, which was in a cross-polarized configuration relative to the incident light. In a similar way, this configuration also allows the instrument to filter out light reflected from the cornea. Conversely, when standard paper was placed behind the target while maintaining the same optical configuration, the target became visible in transmission. The paper acts as a diffuse scatterer (analogous to the sclera at the back of the eye). When the light passes through the target and strikes the paper, the latter randomizes the polarization of this incident light. A portion of this depolarized light then can travel back, illuminate the target in transmission and pass through the beam splitter to be detected by the camera.

Fig. 2. a, Tabletop clinical research prototype with a computer. b,c, Illustration of the instrument's capability to suppress reflected light while detecting transmitted signals. A 1951 USAF target (R1DS1P1, Thorlabs, USA) was used as a sample. Both transmission and reflection tests were conducted using the same low optical power (a few mW).

Reviewer 2:

The authors have developed and tested an affordable microscope for corneal imaging. A key achievement is the modification of the method described in “In vivo corneal and lenticular microscopy with asymmetric fundus retroillumination” by Weber and Mertz, allowing them to obtain high-resolution images with a significantly larger field of view. This innovation could enable better screening in resource-limited settings.

The work has substantial significance, particularly for ophthalmology and global health applications. By providing a cost-effective solution for corneal imaging, this device has the potential to improve accessibility to eye care in rural and developing areas where corneal infections are more prevalent. Compared to established literature, the device offers a larger field of view, which could be a valuable advantage over existing commercial systems.

While the work demonstrates promising results, some claims—such as the significantly larger viewing area and cost-effectiveness—would benefit from additional references or quantitative comparisons to commercially available systems. Furthermore, some numerical discrepancies in calculations (e.g., NA values, spot size estimations) require clarification. Additional safety considerations for subject’s head alignment (before imaging) and more detailed explanations of equations used in safety assessments would help support the conclusions.

There are no fundamental flaws that would prohibit publication, but several areas require revision:

- Some calculations do not align with expected values and need clarification.
- The methodology for measuring endothelial cell density and nerve density using a neural network requires justification over more traditional segmentation approaches.
- Claims regarding resolution and contrast control require additional experimental validation, such as measuring resolution with a resolution target.

Overall, the methodology appears sound, with a well-thought-out approach to modifying existing techniques.

The manuscript lacks some critical details necessary for full reproducibility. Specifically:

- Several key parameters used in safety calculations are not fully described, making it difficult for readers to verify the approach without direct access to ISO 15004-2:2007 and ANSI Z80.36-2021 standards.
- Additional methodological details, such as how cell diameters were measured and how image acquisition settings impact resolution, would improve transparency.

The manuscript presents an important and promising advancement in affordable corneal imaging. With revisions to improve clarity, address numerical inconsistencies, and provide additional methodological details, the work has the potential to make a significant contribution to the field.

Below, I provide detailed feedback and suggestions for improving the clarity and rigor of your manuscript. I hope these comments will help refine your work and enhance its presentation:

We thank the reviewer for the insightful and detailed comments, all of which are addressed step-by-step below, with the corresponding changes implemented directly in the manuscript text.

1. Lines 85-86: The sentence appears unclear. I recommend rewriting it for better clarity.

The original sentence was: 'The insightful visualization of the above phenomenon comes from a simulation of the wave equation in Figs.1e-i.'

The sentence was rewritten for clarity:

"The emergence of interference phenomena in transmission can be modeled through simulations of the wave equation, as presented in Figs. 1e–i. See the simulation details in *Methods*."

2. Line 68: Please replace "focus at" with "focus on" for correct phrasing.

Done.

3. Lines 85-101: The terms "first lens" and "second lens" in the description of the setup are vague. Consider assigning specific names to the optical elements and using those consistently when describing the setup and experiments.

Indeed, the text was improved following the assigning of specific names:

"We modelled the transmission as a 4F system with two lenses (objective lens and tube lens) that conjugate the sample to the camera plane. For simplicity, a single spherical scatterer in the sample is illuminated from below with a quasi-collimated wave of incoherent light. In the absence of the scatterer, the quasi-collimated wave is simply relayed by the 4F optical system, producing a uniform illumination on the camera. In contrast, when the scatterer is placed near the focal plane of the objective lens, a portion of the quasi-collimated wave close to the scatterer is diffracted (diffracted wave), while another portion of the wave located further from the scatterer propagates without diffraction (transmitted wave). Although the two waves propagate within the same optical path they exhibit different angular divergences – the diffracted wave fills a significantly larger portion of the NA of the objective lens. This leads to a spatial separation of the waves in the back-focal plane of the objective lens, where the transmitted wave is focused, and the diffracted wave is collimated (see the zoomed Figs. 1e,h and *Supplementary Video 1*). The tube lens recombines both waves on the camera, producing dark or bright interference contrast for the scatterer."

4. Lines 98-99: In the sentence "The sign of the contrast, denoting constructive or destructive interference, can be alternated by moving the camera, the scatterer, or the lenses," please specify the direction and axis of movement.

The sentence was modified accordingly:

"The sign of the contrast, denoting constructive or destructive interference, can be alternated by moving axially the camera, the scatterer, or the lenses (as shown with the light blue arrows in Figs. 1g,i)."

We have also added details to the Fig. 1 caption:

"The axial shift of the scatterer around the focal plane (see the light blue arrows) changes its visible brightness in the camera plane, in agreement with the experiment."

5. Line 138: The refractive index of the cornea-lens ($n_{\text{cornea-lens}}$) is missing.

We have added the value and the corresponding reference:

"... where $n_{\text{cornea-lens}}$ is approximated as 1.4³⁶."

[36] Optical Properties of the Eye - American Academy of Ophthalmology. <https://www.aao.org/education/munnerlyn-laser-surgery-center/optical-properties-of-eye>.

6. Line 140: The refractive index in the anterior eye (n) is also not provided.

For clarity, we now use only one refractive index $n_{\text{cornea-lens}}$ throughout the text.

7. Lines 141-143: How were the NA_i values calculated? Assuming the only variable is the source radius, the ratio of the two NA_i values should equal $15/0.3 = 50$, but in your manuscript, it equals 40. Could you clarify this discrepancy?

Indeed, the numerical aperture is directly proportional to the source size $NA_i \approx n_{\text{cornea-lens}} \cdot \frac{\text{Source radius}}{\text{Eye length}}$

only for the case of small angles. For large angles (in particular the case of 15 mm source), one needs to use the trigonometric relation

For small 0.3 mm source: $NA_i = n_{\text{cornea-lens}} \cdot \sin\left(\arctan\left(\frac{3\text{ mm}/2}{23\text{ mm}}\right)\right) = 0.0091 \approx 0.01$

For large 15 mm source: $NA_i = n_{\text{cornea-lens}} \cdot \sin\left(\arctan\left(\frac{15\text{ mm}/2}{23\text{ mm}}\right)\right) = 0.4340 \approx 0.43$, which gave 0.4

on the rounding, as written in the main text.

We have replaced the formula by $NA_i = n_{\text{cornea-lens}} \cdot \sin\left[\arctan\left(\frac{\text{Source radius}}{\text{Eye length}}\right)\right]$. We have also replaced the

"=" sign with the approximation "≈" sign and rounded values to two decimal places:

"Indeed, experimentally, the scatterers illuminated with a small source (0.3 mm) and low NA_i ($NA_i \approx 0.01$) experience a smaller NA_{eff} and therefore fade at a slower rate with defocus than the same scatterers illuminated with a large source (15 mm) and high NA_i ($NA_i \approx 0.43$) (Fig. 3b)."

8. Line 148: The phrase "light of higher coherence" is ambiguous. Do you mean spatial coherence? The same applies to Line 282: "controlling the coherence of light." Please specify the type of coherence.

Indeed, we have now replaced 'coherence' by 'spatial coherence' in both instances.

9. Lines 155-156: The enlarged spot size of 2.4 mm (reported in Methods) does not match the calculated value based on the equation in Line 400: $\sqrt{1.7^2 + 2 \times 1 \times 0.33} = 1.88$ mm. Similarly, the value of 3.4 mm in Line 160 could not be reproduced (I calculated 3.1 mm). Could you explain these differences?

In the equation $D=2 \cdot \sigma_{total} = 2 \cdot \sqrt{\sigma_0^2 + 2 \cdot z \cdot l^*}$ the σ_0 should be the radius and not the diameter of the initial beam. Indeed, at the surface, where $z = 0$, we get $D = 2 \sigma_0 = 1.7$ mm, as expected.

Then, $D=2 \cdot \sigma_{total} = 2 \cdot \sqrt{\sigma_0^2 + 2 \cdot z \cdot l^*} = 2 \cdot \sqrt{(1.7/2)^2 + 2 \cdot 1 \cdot 0.33^*} \approx 2.4$ mm, in agreement with the manuscript.

Similarly, $D=2 \cdot \sigma_{total} = 2 \cdot \sqrt{\sigma_0^2 + 2 \cdot z \cdot l^*} = 2 \cdot \sqrt{(3/2)^2 + 2 \cdot 1 \cdot 0.33^*} \approx 3.4$ mm

This point is now additionally clarified in the text:

“Considering forward and backward propagation through the 1 mm thick sclera the increased spot size is given by: $D=2 \cdot \sigma_{total} = 2 \cdot \sqrt{\sigma_0^2 + 2 \cdot z \cdot l^*}$, where σ_{total} is the total variance, σ_0 is the initial spot radius and z is 1 mm sclera thickness.”

10. Lines 184-188: This section would benefit from a summary of the results presented in the supplementary materials to tie them back to the main text.

Thanks to the formatting options available in *Nature Communications*, we were able to move several supplementary figures into the main text:

“Reduced pupil size and increased scattering of the crystalline lens (cataract) are common among a large population of elderly subjects. To study their effects on interference contrast, we used an artificial model eye having replaceable anterior part featuring either dilated/contracted pupil or transparent/scattering lens (Fig. 4).

The pupil size determines the width of the quasi-collimated illumination beam injected into the eye and, therefore, directly affects the amount of detected light on the camera (Fig. 4b). Although it is optically feasible to produce a narrow illumination beam that would not be partially blocked by the pupil, this implementation comes at the cost of reduced contrast, either due to increased source magnification or less efficient light collection from the LED. Consequently, in experiments involving human subjects, we opted for a more direct solution: imaging in a dark environment to promote natural pupil dilation and maximize effective illumination.

The scattering of the crystalline lens causes the illumination to diffuse across a larger area of the sclera, thereby increasing the NA_i of the secondary illumination. This leads to a reduction in interference contrast but also results in a shallower DOF, which improves resolution (Fig. 4c). A similar effect is also produced by high ocular aberrations.”

[editorial note: panel redacted]

Fig. 4. **a**, Full device from the experiments with the artificial eye. **b**, Effect of pupil size on the signal. **c**, Influence of crystalline lens scattering (cataract) on the interference contrast and on DOF/resolution.

11. Line 203: How were cell diameters measured? I could not find this information in the Methods section.

We have added the new part ‘Quantitative image analysis’ to the methods and have explained:

“Characteristic cell sizes in the images were quantified by manually measuring diameters of several cells in Fiji⁶⁴ and calculating the mean.”

12. Lines 207-208: Why did you use a U-net neural network for nerve density evaluation rather than a more classical segmentation approach?

While the manual segmentation is sufficient for determining nerve density in the image of the manuscript, this approach is not scalable for future clinical trials. In particular, manual segmentation of one transmission interferometric image can take 5× longer than CM, because of larger FOV. Neural network segmentation is an important solution for automating this process.

We have added the clarification in the main text:

“The implementation of neural networks for segmentation is particularly valuable in the context of future clinical trials, where the combination of large amounts of data and the extended field-of-view provided by the transmission method would render manual segmentation impractical.”

13. Lines 211-212: The claim of achieving a "10× larger viewing area" needs a reference. Additionally, the HRT3 device by Heidelberg Engineering reportedly has a field of view of 400 × 400 μm, which is not 10 times smaller than the reported 1 mm field of view (source: <https://business-lounge.heidelbergengineering.com/ie/en/products/hrt3-rcm/hrt3-rcm/publications/>) .

The confusion arises from the comparison of different metrics: field-of-view and area. When comparing field-of-view, we indeed observe a smaller 2 mm / 0.4 mm = 5× improvement over the confocal method.

We have standardized the text to ensure that comparisons are made using field-of-view only. The sizes of the state-of-the-art SM and CM have been added to the text with the references:

“Moreover, new transmission design gives access to 2 mm x 1 mm FOV, which is about 5× larger than the SM (0.25 mm × 0.55 mm)⁴⁷ and CM (0.4 mm × 0.4 mm)⁴⁸ in clinics. “

[47] CEM-530 | Technical Specifications | NIDEK USA. <https://usa.nidek.com/cem-530/specifications/> (2024).

[48] HRT3 RCM - In vivo corneal confocal microscope | Heidelberg Engineering. <https://business-lounge.heidelbergengineering.com/us/en/products/hrt3-rcm/hrt3-rcm/>.

14. Lines 215-216: How was the endothelial cell density measured?

The cells were counted manually using the Point Tool in Fiji, and their number was then divided by the area within which they were counted. It is important to note that the cells could appear with either bright or dark interference contrast, depending on their axial position relative to the optical focus. Due to the natural curvature of the cornea, both contrast types could be present within the same image. To avoid ambiguity, only the bright cells were counted, and their number was divided by the corresponding observation area.

Additional Fig. 2. Manually counted bright endothelial cells.

We added clarifications in the main text:

“The cells were visible with bright or dark interference contrast, depending on their axial location relative to the optical focus. Due to the natural corneal curvature, both contrasts could be observed within the same image. Importantly, in the exact optical focus, the phase difference between the transmitted and diffracted waves is minimal, rendering the cells invisible. To avoid ambiguity, cell density was calculated by counting only the bright cells and dividing their number by the area in which they were observed.”

Additional clarifications in the methods:

“The density of cells was calculated by counting the cells using Point tool of Fiji⁶⁴ and dividing their number by the area in which they were observed.”

15. Lines 225-226: The statement “16× larger viewing area than the state-of-the-art SM” lacks a reference.

In agreement with the above question 13, we have adapted the comparison metric (area -> field of view) and provided the corresponding references:

“Moreover, new transmission design gives access to 2 mm x 1 mm FOV, which is about 5× larger than the SM (0.25 mm × 0.55 mm)⁴⁷ and CM (0.4 mm × 0.4 mm)⁴⁸ in clinics.”

[47] CEM-530 | Technical Specifications | NIDEK USA. <https://usa.nidek.com/cem-530/specifications/> (2024).

[48] HRT3 RCM - In vivo corneal confocal microscope | Heidelberg Engineering. <https://business-lounge.heidelbergengineering.com/us/en/products/hrt3-rcm/hrt3-rcm/>.

16. Figures 4 and 5: The green text is difficult to read. I suggest changing the font color for better visibility.

Indeed, the green text was made larger and brighter in both figures.

17. Figure 4e: Why are images from a different subject shown? It would be preferable to show images from the same subject and location. If this is not possible, please provide detailed information about the "other subject" and the setup used.

Following the reviewer's recommendation, we performed a specular microscopy (SM) exam on the same FED patient. As a result, the images shown in Figure 4d (transmission method) and Figure 4e (SM) now originate from the same subject. However, it is important to note that the SM image could not be acquired from the same central region of the cornea. Specifically, due to corneal edema caused by FED and the associated increase in corneal haze, the SM image from the central cornea was saturated (see Additional Fig. 3 below). To enable a meaningful comparison with the transmission results, the SM image was instead acquired from a paracentral region.

Additional Fig. 3. Specular microscopy images from different corneal regions of the same FED subject.

We have added the SM image as well as the information to the figure caption:

. e, Comparison with specular microscopy from the same subject, but a different (paracentral) corneal region. The central SM image was saturated due to substantial corneal haze....

18. Lines 275-276: How was the cell density measured?

The density of crystalline lens epithelial cells was measured similarly to that of endothelial cells—by counting the cells using the Point Tool in Fiji⁶⁴ and dividing the count by the area in which they were observed.

We have added a link to the Methods section in the main text, as well as a corresponding paragraph within the Methods section itself:

“The cells were packed with a density of about 5000 cells/mm² (see the measurement details in *Methods*).”

“The density of cells was calculated by counting the cells using Point tool of Fiji⁶⁴ and dividing their number by the area in which they were observed.”

19. Lines 299-301: Could you provide an estimated cost of the device and specify how it compares to commercially available systems?

While the price of a commercial clinical confocal microscope is known to be around €60,000, a direct comparison is difficult, as this price includes not only the optical engine but also labor, CE/FDA certification, and other associated expenses. Similarly, the cost of our complete instrument is not fully representative, as it is expected to decrease significantly with mass production—for instance, through the replacement of costly research-grade mechanical parts and the substitution of Thorlabs/Edmund Optics components with more cost-effective alternatives.

Nevertheless, the cost of the two most essential and expensive components—the camera and the microscope objective—is expected to remain relatively constant. For this reason, we provide the costs of these components in the text:

“Additionally, given the relatively low cost of its instrumentation - where the most expensive components are a consumer-grade USB camera (under \$1000) and a microscope objective (under \$3000) - and the robustness of its common-path interferometric design, this device presents opportunities for screening the population in the rural areas of developing countries, where infections are most prevalent.”

20. Line 333: The eye focal length value requires a reference and the name of the eye model used.

In line 333 we referred to in vivo human eye. We have added the reference to the typical eye focal length:

“Considering the common 17 mm value for the focal length of the eye⁶⁴, the spot size was $1.8 \text{ mm} \times 17 \text{ mm}/18 \text{ mm} = 1.7 \text{ mm}$ for an objective with 18 mm FL and $1.8 \text{ mm} \times 10 \text{ mm}/18 \text{ mm} = 3 \text{ mm}$ for an objective with 10 mm FL.”

[64] Walker, B. H. *Optical Design for Visual Systems*. (SPIE Press, 2000).

Later, in line 375 we discussed the artificial model eye, used in some experiments, and have already provided its name:

“We used an artificial eye (Model Eye for OCT, Modell-Augen Manufaktur, Germany) having a modular structure.”

21. Lines 334-335: For a 10 mm focal length objective, the calculation appears incorrect. Should it be $1.8 \text{ mm} \times 17 \text{ mm} / 10 \text{ mm} = 3.06 \text{ mm}$?

Thank you for pointing out the typo; the formula has now been corrected:

“... and $1.8 \text{ mm} \times 40 \text{ mm}/40 \text{ mm} \times 17 \text{ mm}/10 \text{ mm} = 3 \text{ mm}$ for an objective with 10 mm FL. Here, 40 mm refers to the focal length of the lenses used in a 1:1 magnification lens pair.”

22. Lines 338-339: It would be helpful for readers to include information about the sensor pixel size in the manuscript.

The requested information has been added:

“The ASI432 featured a global shutter with 1608×1104 pixels and a pixel size of 9 microns. It had increased imaging speed of 120 frames/s (fps), which was beneficial for suppressing artefacts related to eye movement. The ASI585MC, on the other hand, had a rolling shutter sensor and a slower 45 fps rate, but it featured 3840×2160 pixels (pixel size $2.9 \mu\text{m}$), allowing for an extended FOV.”

23. Line 339: Replace “Using a camera with high QE ...” with “Using a camera with a high quantum efficiency (QE) ...” for clarity.

Done.

24. Lines 343-344: What kind of eye movements are you referring to? Please elaborate.

We have added the well-known types of ocular motion that affect high-resolution ophthalmic systems:

“It had an increased imaging speed of 120 frames per second (fps), which was beneficial for suppressing artefacts related to eye movements, including those caused by heartbeat, breathing, and fixational eye motions.”

25. Line 351: If you are providing manufacturer details for other components, the corresponding information here is missing.

We have added the information:

“An additionally implemented linear polarizer (LPNIRE100-B, Thorlabs, USA) was necessary to produce a cleaner polarization state in the illumination.”

26. Line 367: Where does the value of the lens refractive index originate?

We have added the clarifications:

“Refractive indexes of the medium and lenses were 1 (corresponding to air) and 1.5 (corresponding to glass), respectively.”

27. Lines 375-378: According to the manufacturer’s website, the light-scattering lens imitates a patient’s natural lens, not a cataract lens (source: <https://modell-auge-manufaktur.de/wp-content/uploads/2023/06/brochure-OCT-model-eye.pdf>). Did you use a different, more scattering lens? Please clarify.

Although it is true that the brochure presents the light-scattering lens as a natural lens, we noticed that it exhibits stronger scattering properties than expected in a young healthy eye. This is highlighted in Additional Fig. 4 below, which compares the OCT scan of the scattering lens presented in the brochure [1] with a typical OCT scan of a healthy human eye [2]. The strong scattering of the lens is also evident in our macroscopic view in Figure 4C, where the letters behind the lens become unreadable due to pronounced diffusive scattering.

[1] <https://modell-auge-manufaktur.de/wp-content/uploads/2023/06/brochure-OCT-model-eye.pdf>

[2] Ang, M. *et al.* (2018) ‘Anterior segment optical coherence tomography’, *Progress in Retinal and Eye Research*, 66, pp. 132–156. doi:10.1016/j.preteyeres.2018.04.002.

Artificial eye with a scattering lens

Normal in vivo eye

Additional Fig. 4. Comparison of OCT scans of the artificial eye with a scattering lens (taken from the brochure [1]) and a normal human in vivo eye [2]. The scattering signal from the artificial eye is noticeably stronger.

Fragment of Figure 4C showing the university logo viewed through the transparent and scattering lenses. The scattering lens exhibits considerable diffusive blurring.

We have added to the figure 4 caption:

“... The strong scattering of the lens is evident in the macroscopic view through the artificial eye, where the letters behind the lens become unreadable due to pronounced diffusive scattering.”

28. Lines 420-434: The safety considerations outlined in this section rely on ISO 15004-2:2007 and ANSI Z80.36-2021 standards, which may not be readily accessible to all readers. To improve clarity and reproducibility, I suggest elaborating further on the equations and values used in your analysis. Providing additional details would help readers follow and verify your approach without requiring direct access to these standards.

It proves very challenging to condense the equation explanations dispersed throughout the 40-page standard into the article without omitting critical details. **Nevertheless, we have made our best effort to provide additional clarifications below, which are included in the new Supplementary file.**

The reference to the Supplementary file is added to the Methods:

‘For the extended light exposure calculations see Supplementary file.’

Calculations in Supplementary file:

We will focus on ISO 15004-2:2007, which is the main ophthalmic standard used in France.

Our device employs pulsed near-infrared (NIR) illumination, making it subject to the exposure limits defined in 5.5.2.1 and 5.5.2.2 of Table 6 in the ISO standard. According to the standard:

FOR RETINA:

$$5.5.2.1: H_{\text{VIR-R}} = \sum_{\lambda=380}^{1400} (E_{\lambda} \times \Delta t) \times R(\lambda) \times \Delta \lambda < \left(\frac{10}{d_r} t^{3/4} \right) \frac{\text{J}}{\text{cm}^2}, \text{ where } E_{\lambda} \text{ is spectral irradiance at}$$

wavelength λ , t is a time within which the safety of the instrument is evaluated (applicable, when equal or lower than 20 s - see details below), Δt is the total duration of pulses within time t (both times measured in seconds), $R(\lambda)$ is visible and infrared thermal hazard weighting function (determined by a Table A.1 in Annex), $\Delta \lambda$ wavelength summation interval, d_r is the minimum retinal image diameter of the source (expressed in millimetres), limited to the range 0.03–1.7 mm.

Moreover, for repetitive pulses, the above retinal limit should be further reduced by a factor of $N^{-1/4}$, where N is the number of pulses within the evaluated time exposure period. Although this clause is formally defined for lasers, we conservatively apply it to our LED source for additional security.

The smallest illuminated retinal area produced by our instrument is 1.7 mm. The maximum optical power entering the eye is 100 mW, which results in $E = \frac{100 \text{ mW}}{(1.7 \text{ mm})^2} = 3.46 \frac{\text{W}}{\text{cm}^2}$ irradiance per pulse.

In this calculation, we have accounted for the fact that the LED chip has a square spatial profile, rather than a circular one. The LED spectrum is centered at 850 nm, with a bandwidth ranging from 800 to 875 nm. For the sake of safety evaluation, we assume a worst-case scenario in which all emission occurs at the most hazardous wavelength of 800 nm—this corresponds to the thermal hazard weighting function $R(800 \text{ nm}) = 0.63$. Additionally, we simplify the safety equation by assuming that the pulses are uniform and rectangular in shape. Under these assumptions, the general formula reduces to:

$$H_{\text{VIR-R}} = E \times (N \times t_{\text{pulse}}) \times R(800 \text{ nm}) < \left(\frac{10}{d_r} t^{3/4} \right) \frac{\text{J}}{\text{cm}^2} \times N^{-1/4}, \text{ where } E \text{ is irradiance per pulse at}$$

800 nm, $(N \times t_{\text{pulse}})$ is the total duration of pulses within time t with N being the number of pulses within time t and t_{pulse} being the duration of the pulse.

For pulsed instruments, the safety should be evaluated for all times within 20 s period. For exposure times greater than 20 s, the limits of the continuous wave instruments apply (will be tested in the section below).

There are three important time points that we evaluate: $t = t_{\text{pulse}} = 10 \text{ ms}$ (single pulse), $t = 110 \text{ ms}$ (two pulses), $t = 20 \text{ s}$ (maximal period of applicability of 5.5.2.1). Calculations for those cases are provided here:

For single pulse ($t = t_{\text{pulse}} = 10 \text{ ms}$):

$$H_{\text{VIR-R}} = 3.46 \frac{W}{\text{cm}^2} \times (1 \times 0.01s) \times 0.63 < \left(\frac{10}{1.7} (0.01s)^{3/4} \right) \frac{J}{\text{cm}^2} \times 1$$

$$0.02 \frac{J}{\text{cm}^2} < 0.18 \frac{J}{\text{cm}^2}, \text{ which is } 9\times \text{ below the safety limit.}$$

For two pulses ($t=110 \text{ ms}$):

$$H_{\text{VIR-R}} = 3.46 \frac{W}{\text{cm}^2} \times (2 \times 0.01s) \times 0.63 < \left(\frac{10}{1.7} (0.11s)^{3/4} \right) \frac{J}{\text{cm}^2} \times (2)^{-1/4}$$

$$0.05 \frac{J}{\text{cm}^2} < 0.94 \frac{J}{\text{cm}^2}, \text{ which is } 18\times \text{ below the limit.}$$

For maximal period of applicability of 5.5.2.1 ($t = 20 \text{ s}$):

$$\text{Number of pulses within 20 seconds: } N = \frac{20s}{0.01s + 0.90s} = 200$$

$$H_{\text{VIR-R}} = 3.46 \frac{W}{\text{cm}^2} \times (200 \times 0.01s) \times 0.63 < \left(\frac{10}{1.7} (20s)^{3/4} \right) \frac{J}{\text{cm}^2} \times (200)^{-1/4}$$

$$4.4 \frac{J}{\text{cm}^2} < 14.7 \frac{J}{\text{cm}^2}, \text{ which is } 3\times \text{ below the safety limit.}$$

For durations longer than 20 seconds, we apply the continuous wave standard (5.5.1.5 in Table 4).

$$\mathbf{5.5.1.5:} \quad E_{\text{VIR-R}} = \sum_{\lambda=380}^{1400} E_{\lambda} \times R(\lambda) \times \Delta\lambda < \left(\frac{1.2}{d_r} \right) \frac{W}{\text{cm}^2}$$

Here we should consider our instrument as having continuous emission but with a time-averaged irradiance of $E_{\text{avg}} = E_{\text{pulse}} \times \frac{0.01s}{0.01s + 0.09s} = 3.46 \frac{W}{\text{cm}^2} \times 0.1 = 0.35 \frac{W}{\text{cm}^2}$. We will keep considering that all emission occurs on the most limiting wavelength of 800 nm. Then:

$$E_{\text{VIR-R}} = E_{\text{avg}} \times R(800 \text{ nm}) < \left(\frac{1.2}{d_r} \right) \frac{W}{\text{cm}^2}$$

$$0.35 \frac{mW}{\text{cm}^2} \times 0.63 < \left(\frac{1.2}{1.7} \right) \frac{W}{\text{cm}^2}$$

$$0.22 \frac{mW}{\text{cm}^2} < 0.7 \frac{W}{\text{cm}^2}, \text{ which is more than } 3\times \text{ below the safety limit.}$$

In summary of the above calculations, the instrument operates at a level 3× below the retinal safety limit.

We now provide a similar set of extended safety calculations for the cornea/anterior eye. For pulsed instruments, the relevant exposure limits are defined in 5.5.2.2 and 5.5.2.3 of the Table 6 of the ISO

standard. Among these, criteria 5.5.2.2 is more restrictive within our spectral range and is therefore used for the evaluation.

FOR CORNEA:

5.5.2.2: $H_{\text{IR-CL}} = \sum_{\lambda=770}^{2500} H_{\lambda} \times \Delta\lambda > 1.8 \times t^{1/4} \frac{\text{J}}{\text{cm}^2}$, where H_{λ} is the radiant exposure per wavelength

measured in $\frac{\text{J}}{\text{cm}^2}$, t is a time within which the safety of the instrument is evaluated (applicable, when equal or lower than 20 s).

The radiant exposure was calculated based on irradiance measurements performed in accordance with the safety standard protocol, using a 0.9 mm averaging aperture. A powermeter (S130C, Thorlabs, USA), covered with a plate featuring a 0.9 mm through-hole (CPA1, Thorlabs, USA), measured an optical power of 3 mW. This corresponds to corneal irradiance of $E = \frac{3 \text{ mW}}{\pi/4 \times (0.9 \text{ mm})^2} = 0.5 \frac{\text{W}}{\text{cm}^2}$.

Considering that all emission occurs at the same wavelength (directly applicable because of the absence of weighting function) as well as the uniform pattern of pulses and their rectangular shape, the equation simplifies to:

$H_{\text{IR-CL}} = E \times (N \times \Delta t) < 1.8 \times t^{1/4} \frac{\text{J}}{\text{cm}^2}$, where $(N \times \Delta t)$ is the total duration of N pulses having width Δt within time t (both times measured in seconds).

For single pulse: $t = t_{\text{pulse}} = 10 \text{ ms}$:

$$0.5 \frac{\text{W}}{\text{cm}^2} \times (1 \times 0.01 \text{ s}) < 1.8 \times (0.01 \text{ s})^{1/4} \frac{\text{J}}{\text{cm}^2}$$

$$0.005 \frac{\text{J}}{\text{cm}^2} < 0.57 \frac{\text{J}}{\text{cm}^2}, \text{ which is more than } 100\times \text{ below the limit.}$$

For two pulses: $t = t_{\text{pulse}} = 110 \text{ ms}$:

$$0.5 \frac{\text{W}}{\text{cm}^2} \times (2 \times 0.01 \text{ s}) < 1.8 \times (0.110 \text{ s})^{1/4} \frac{\text{J}}{\text{cm}^2}$$

$$0.01 \frac{\text{J}}{\text{cm}^2} < 1.03 \frac{\text{J}}{\text{cm}^2}, \text{ which is } 100\times \text{ below the limit.}$$

For maximal period of applicability of 5.5.2.2 ($t = 20 \text{ s}$):

Number of pulses within 20 seconds: $N = \frac{20 \text{ s}}{0.01 \text{ s} + 0.90 \text{ s}} = 200$

$$0.5 \frac{\text{W}}{\text{cm}^2} \times (200 \times 0.01 \text{ s}) < 1.8 \times (20 \text{ s})^{1/4} \frac{\text{J}}{\text{cm}^2}$$

$$1 \frac{J}{cm^2} < 3.8 \frac{J}{cm^2}, \text{ which is more than } 3\times \text{ below the limit.}$$

For durations longer than 20 seconds, we apply the continuous wave standard (5.5.1.3 in Table 4).

$$\mathbf{5.5.1.3:} E_{IR-CL} = \sum_{\lambda=770}^{2500} E_{\lambda} \times \Delta\lambda < 100 \frac{mW}{cm^2}$$

Here, we consider our instrument as having continuous emission, but with a time-averaged irradiance of: $E_{avg} = E_{pulse} \times \frac{0.01s}{0.01s + 0.09s} = 0.5 \frac{W}{cm^2} \times 0.1 = 50 \frac{mW}{cm^2}$. We will keep considering that all emission occurs on the single wavelength. Then:

$$50 \frac{mW}{cm^2} < 100 \frac{mW}{cm^2}, \text{ which is } 2\times \text{ below the limit.}$$

In summary of the above calculations, the instrument operates at 2× below the safety limit for the cornea and 3× below the limit for the retina.

The above evaluations were based on the most commonly used illumination mode, consisting of a 10 ms pulse followed by a 90 ms break. In an alternative mode, the eye can be illuminated with a 1-second pulse followed by a 19-second break. This mode is particularly useful for acquiring fly-through volume stacks without temporal delay between the consecutive frames. Repeating the same calculations for this alternative mode:

5.5.2.1 (retinal safety, pulsed):

$$H_{VIR-R} = E \times (N \times t_{pulse}) \times R(800nm) < \left(\frac{10}{d_r} t^{3/4} \right) \frac{J}{cm^2} \times N^{-1/4}$$

$$H_{VIR-R} = 3.46 \frac{W}{cm^2} \times (1 \times 1s) \times 0.63 < \left(\frac{10}{1.7} (1s)^{3/4} \right) \frac{J}{cm^2} \times 1$$

$$2.2 \frac{J}{cm^2} < 5.8 \frac{J}{cm^2}, \text{ which is more than } 2\times \text{ below the safety limit.}$$

5.5.1.5 (retinal safety, time-averaged):

$$E_{VIR-R} = E_{avg} \times R(800nm) < \left(\frac{1.2}{d_r} \right) \frac{W}{cm^2}, \text{ where:}$$

$$E_{avg} = E_{pulse} \times \frac{1s}{1s + 19s} = 3.46 \frac{W}{cm^2} \times 0.05 = 0.17 \frac{W}{cm^2}$$

Then:

$$0.17 \frac{W}{cm^2} \times 0.63 < \left(\frac{1.2}{1.7} \right) \frac{W}{cm^2}$$

$$170 \frac{mW}{cm^2} < 700 \frac{mW}{cm^2}, \text{ which is } 4\times \text{ below the safety limit.}$$

5.5.2.2 (corneal safety, pulsed):

$$H_{\text{IR-CL}} = E \times (N \times \Delta t) < 1.8 \times t^{1/4} \frac{\text{J}}{\text{cm}^2}$$

$$0.5 \frac{\text{W}}{\text{cm}^2} \times (1 \times 1 \text{ s}) < 1.8 \times (1 \text{ s})^{1/4} \frac{\text{J}}{\text{cm}^2}$$

$$0.5 \frac{\text{J}}{\text{cm}^2} < 1.8 \frac{\text{J}}{\text{cm}^2}, \text{ which is more than } 3\times \text{ below the limit.}$$

5.5.1.3 (corneal safety, time-averaged):

$$E_{\text{IR-CL}} = \sum_{\lambda=770}^{2500} E_{\lambda} \times \Delta\lambda < 100 \frac{\text{mW}}{\text{cm}^2}$$

$$E_{\text{avg}} = E_{\text{pulse}} \times \frac{1 \text{ s}}{1 \text{ s} + 19 \text{ s}} = 0.5 \frac{\text{W}}{\text{cm}^2} \times 0.05 = 25 \frac{\text{mW}}{\text{cm}^2}$$

$$25 \frac{\text{mW}}{\text{cm}^2} < 100 \frac{\text{mW}}{\text{cm}^2}, \text{ which is } 4\times \text{ below the limit.}$$

As a result, the 1-second pulse – 19-second break illumination mode remains within safety limits, operating at 3× below the maximum permissible exposure for the cornea and 2× below the limit for the retina.

Reviewer 3:

Imaging the anterior segment of the eye with cellular-level details remains a technological challenge. In this paper, the authors proposed a transmission interference microscopy technique in which the backscattered light from the sclera serves as a secondary source of illumination and imaging of the anterior eye. The image of individual layers of the eye within the focal plane of the microscopic objective was formed by the interference between the backscattered light from cellular features and the transmitted light non-affected by the cellular features. In this work, the authors also evaluated the impact of the size of the secondary source (e.g., the size of the illumination spot on the sclera) on the performance (contrast, resolution, DOF) of the imaging system. The proposed method enabled a 2-mm FOV of the anterior eye with cellular resolution, much wider than that of standard clinical and research systems. The system's performance was demonstrated through imaging of four young healthy subjects, one FED patient with cataract, and one elder subject. The paper is scientifically sounded and appropriate for publication in Nature Communication provided that authors address the concerns below:

We thank the reviewer for the positive assessment of our work and for recognizing the novelty and scientific merit of the proposed transmission interference microscopy method. Below we provide point-by-point responses to all concerns raised, along with the corresponding revisions implemented in the text.

1. The formulas in lines 138-140 indicate that the higher the NA_i , the higher the NA_{eff} , which seems to contradict the experimental observation (e.g., in lines 141-143). It is not clear why the increased contrast of the interference with the size of the illuminated sclera is the result of the NA_i , i.e., the numerical aperture of the illuminated sclera visible to the scatterer in the anterior eye. Considering the sclera as a condenser instead of a scattering element may be a strong assumption. I encourage the authors to explore other alternatives. It is likely that the improved contrast of the interference with the size of the illuminated sclera simply has to do with the spatial coherence of the secondary source instead of the NA_i the way it is defined by the authors. The larger the source, the lower the spatial coherence and the lower the interference contrast due to the scrambling of the phase from different points of the source. This consideration also affects the evaluation of the impact of pupil size on the contrast, as well as the explanation of different scattering behaviors in lines 175-183 (major).

Indeed, a higher NA of the secondary illumination (NA_i) results in a higher NA_{eff} . We also find that this stays in agreement with experimental observation (in lines 141-143). Indeed, when the NA_i is low, the depth-of-field is extended, thus the scatterers remain visible even with significant defocus, in agreement with our text: *'Indeed, experimentally, the scatterers illuminated with a small source (0.3 mm) and small NA_i ($NA_i \approx 0.01$) fade at a slower rate with defocus than the same scatterers illuminated with a large source (15 mm) and large NA_i ($NA_i \approx 0.43$) (Fig. 2b).'* This effect is evident in Fig. 4b, where the smaller source and smaller NA_i leads to more elongated axial PSF profile:

Fragment of Fig. 2b showing that the smaller source leads to more extended axial PSF profile of the scatterer.

Nevertheless, we have added a few clarifications to the text:

'Indeed, experimentally, the scatterers illuminated with a small source (0.3 mm) and low NA_i ($NA_i \approx 0.01$) experience a smaller NA_{eff} and therefore fade at a slower rate with defocus than the same scatterers illuminated with a large source (15 mm) and high NA_i ($NA_i \approx 0.43$) (Fig. 2b). As an additional note, in the latter case the NA_i exceeds the numerical aperture of the microscope objective

$NA_d = 0.3$ (used in that experiment); therefore, only the portion of illumination within the acceptance cone of the objective $NA_i = 0.3$ is utilized.'

Regarding the origin of the increased contrast with the smaller NA and its relation with the spatial coherence - we completely agree with the reviewer! In fact, we have written that in the original text: 'Conversely, with a point source, the scatterer is illuminated by light of higher coherence, originating from a single illumination angle. This results in a more defined phase relationship between the transmitted and diffracted waves, leading to higher contrast.' **For additional clarification, we have now replaced the word 'coherence' with 'spatial coherence' through the main text.**

The interpretations of the phenomena in terms of NA and coherence are equivalent, and both are used depending on the discipline: NA—through control of the condenser aperture—is typically preferred in conventional bright-field transmission microscopy, while coherence is more commonly referenced in reflection-based methods. To remain accessible to both audiences, we have retained both interpretations in the text.

2. Despite the improved performance with the reduced size of the illuminated sclera, care must be taken to prevent retinal damage. Operating the system with an irradiance that is halfway the MPE limit can be risky as several factors including the refractive error of the patient's eye can quickly put the patient at risk of retina phototoxicity (major).

We thank the reviewer for an important question. We will decouple it into two parts: 1) influence of the refractive errors, 2) influence of the subject's age.

1) Refractive errors cause the projected light to become blurred and spread over a larger retinal area compared to the normal condition. This leads to a reduction in retinal irradiance, thereby improving the safety margin of the device.

Let us consider the example of high myopia of refractive type, with a refractive error of -6 diopters (D). The total optical power of the eye becomes $60\text{ D} + 6\text{ D} = 66\text{ D}$, corresponding to an effective focal length of: $1000\text{ mm} / 66 \approx 15\text{ mm}$. At this focal length, the image is sharply formed. Because of the shorter effective focal length, the image size will be $1.8\text{ mm} \times 40\text{ mm} / 40\text{ mm} \times 15\text{ mm} / 18\text{ mm} = 1.5\text{ mm}$, which is smaller compared the 1.7 mm size in the normal human eye. However, the retina is physically located further back—at the 17 mm effective focal length of a normal eye. Therefore, the light rays are no longer focused on the retina, and the defocus leads to the formation of a blur circle. Assuming a pupil diameter of 4 mm , the blur diameter for a single point of light is: $2 \times 4\text{ mm} / 15\text{ mm} \times (17\text{ mm} - 15\text{ mm}) = 1$. Thus, the total spot size on the retina becomes: 1.4 mm (size in the normal eye) $+ 1\text{ mm} = 2.4\text{ mm}$. We will now use this increased spot size to re-evaluate light safety based on the formulas provided in our response to Question 28 from Reviewer 2.

For retinal safety, the highest restrictions were given by the limit **5.5.2.1** (for pulsed 1 s exposure), which was $2\times$ above our exposure level:

$$H_{\text{VIR-R}} = E \times (N \times t_{\text{pulse}}) \times R(800\text{ nm}) < \left(\frac{10}{d_r} t^{3/4} \right) \frac{\text{J}}{\text{cm}^2} \times N^{-1/4}$$

New irradiance is then:

$$E = \frac{100 \text{ mW}}{(2.4 \text{ mm})^2} \approx 1.8 \frac{\text{W}}{\text{cm}^2}$$

$H_{\text{VIR-R}} = 1.8 \frac{\text{W}}{\text{cm}^2} \times (1 \times 1 \text{ s}) \times 0.63 < \left(\frac{10}{1.7} (1 \text{ s})^{3/4} \right) \frac{\text{J}}{\text{cm}^2} \times 1$, here $d_r = 1.7 \text{ mm}$ is kept in the limit as per requirement of the ISO standard for large sources.

$1.1 \frac{\text{J}}{\text{cm}^2} < 5.8 \frac{\text{J}}{\text{cm}^2}$, which is more than 5× below the safety limit – safer than for normal eye.

2) On the other hand, additional precautions must be taken when imaging children. Young eyes often exhibit both a shorter effective focal length (~ 15 mm) and a shorter axial length, which closely matches the eye's focal length. This anatomical configuration means that the projected image is sharply focused directly on the retina, rather than slightly defocused as in myopic adults. Applying the same geometric scaling as before: $1.8 \text{ mm} \times 40 \text{ mm} / 40 \text{ mm} \times 15 \text{ mm} / 18 \text{ mm} = 1.5 \text{ mm}$. In this case, since the retina is located exactly at the focal plane, no defocus blur occurs, and the full optical energy is concentrated within a smaller spot size of 1.5 mm—compared to 1.7 mm in the average adult eye.

Using the same formulas detailed in our response to Question 28 of Reviewer 2, we calculate the radiant exposure for this more focused scenario. For the most restrictive case of a 1-second pulse (as per ISO 15004-2:2007, **5.5.2.1**), the exposure level is:

$$H_{\text{VIR-R}} = E \times (N \times t_{\text{pulse}}) \times R(800 \text{ nm}) < \left(\frac{10}{d_r} t^{3/4} \right) \frac{\text{J}}{\text{cm}^2} \times N^{-1/4}$$

New irradiance is then:

$$E = \frac{100 \text{ mW}}{(1.5 \text{ mm})^2} \approx 4.5 \frac{\text{W}}{\text{cm}^2}$$

$$H_{\text{VIR-R}} = 4.5 \frac{\text{W}}{\text{cm}^2} \times (1 \times 1 \text{ s}) \times 0.63 < \left(\frac{10}{1.5} (1 \text{ s})^{3/4} \right) \frac{\text{J}}{\text{cm}^2} \times 1$$

$2.8 \frac{\text{J}}{\text{cm}^2} < 6.6 \frac{\text{J}}{\text{cm}^2}$, which is 2.4× below the safety limit. This shows a slight reduction in the gap

between the exposure and the limit as for normal eye we had $2.2 \frac{\text{J}}{\text{cm}^2} < 5.8 \frac{\text{J}}{\text{cm}^2}$, which was 2.6× below the safety limit.

We have the above calculations to the *Supplementary file* and also added the comment to the *Methods* section:

'Refractive errors and the age of the subject can have a minor impact on the safety calculations. For instance, refractive errors tend to spread the light over a larger retinal area, which can improve the safety margin of the device. Conversely, in young children (under 10 years old), the safety margin may be slightly reduced due to differences in ocular geometry. Specifically, young children often have a

shorter effective focal length (around 15 mm) and a correspondingly shorter axial eye length. As shown in the *Supplementary file*, this leads to a smaller illumination spot on the retina—1.5 mm instead of 1.7 mm. In the most limiting case of a 1-second exposure followed by a 19-second break, this results in a short-time radiant exposure of 2.8 J/cm², which is 2.4× below the safety limit of 6.6 J/cm²—a slight reduction compared to the 2.6× margin observed in a normal adult eye.’

3. Lines 218-220: The description of guttae characteristics must be supported by additional data. The external 30-100 μm dark halo could well be the result of shadow created by guttae in the sclera illuminated beam (major).

We agree with the reviewer. Currently, we are in process of collecting the comparison dataset (transmission / specular microscopy) in a cohort of subjects as part of the clinical trial at 15-20 Hospital in Paris that will be able to support this conclusion. **At present, however, this data is not yet accessible to us, therefore we have removed the description of guttae characteristics from the manuscript text:**

The characteristic guttae of FED were visible (Fig. 6d). Moreover, the guttae exhibited dark-bright shifts across the focus further confirming that they are primarily ‘phase’ and not light absorbing objects. In comparison, reflection-based SM depicts guttae as dark spots (Fig. 6e). The characteristic guttae of FED were visible and appeared to be composed of three zones: external 30-100 μm dark halo, 10-20 μm black rings, and 4-8 μm central brighter spot inside them (Fig. 6d). Given that the intensity in transmission microscopy is directly linked to the phase, this suggests a non-uniform internal structure of the guttae in terms of thickness and/or refractive index.

We have also adapted the Fig. 6 by removing the arrows pointing to different ‘parts’ of guttae.

4. Since the imaging performance relies on the transmission of the eye, it will be important to evaluate the utility of the system in disease conditions such as cataracts as well as in highly aberrated eyes (minor).

Indeed, in the original text, Supplementary Fig. 2c illustrated that cataract leads to a reduction in interference contrast, accompanied by an increase in lateral resolution. Additionally, as the subject with Fuchs’ endothelial dystrophy (FED) had a clinically confirmed cataract and was scheduled for cataract surgery, the feasibility of imaging such patients was preliminarily confirmed.

Refractive errors increase the spot size at the back of the eye (as discussed in our response to Question 2), which in turn increases the NA_i of the secondary illumination reaching the anterior eye. This results in a reduction in interference contrast but improves resolution and sectioning. To date, we have successfully imaged subjects with mild refractive errors, while the feasibility of imaging eyes with high aberrations will be further evaluated in our ongoing clinical study at 15–20 Hospital.

To avoid ambiguity and improve clarity, we have now moved the relevant supplementary figures into the main text.

“The scattering of the crystalline lens causes the illumination to diffuse across a larger area of the sclera, thereby increasing the NA_i of the secondary illumination. This leads to a reduction in interference contrast but also results in a shallower DOF, which improves resolution (Fig. 4c). A similar effect is also produced by high ocular aberrations. “

[editorial note: panel redacted]

Fig. 4. **a**, Full device from the experiments with the artificial eye. **b**, Effect of pupil size on the signal. **c**, Influence of crystalline lens scattering (cataract) on the interference contrast and on DOF/resolution.

5. Images have different sizes and it is not specified on what subject such or such images (e.g., figures 3, 6) were collected, nor whether images were from the same imaging session. Please provide images with similar sizes if possible for consistency. Otherwise, clarify the discrepancy in image size (minor).

The differences in image sizes arise from three main factors: 1) variations in the imaging configuration, such as the use of different cameras (e.g., ZWO432 vs. ZWO585) or objectives with different numerical apertures (e.g., 0.3 NA vs. 0.4 NA), and 2) cropping performed to fit images within the limited space of a figure while highlighting the most essential features (for example, the stromal nerve in Fig. 6).

For each image, we have specified whether it was acquired from a healthy subject or a specific patient. Images from different healthy subjects showed only minor variations; therefore, we intentionally avoided identifying individual subjects to maintain clarity and avoid complicating the message of this initial study. A detailed comparison across different healthy subjects, imaging sessions, and clinical operators is part of the ongoing clinical study at 15–20 Hospital in Paris. These broader comparisons are beyond the scope—and length limitations—of the current manuscript, which is focused on presenting the first demonstration of a novel eye imaging method that introduces a new contrast mechanism.

We have added further clarifications in the image captions where subject or device information was previously incomplete:

Fig. 5. Anterior cornea of healthy human subjects in vivo. **a**, Corneal epithelium imaged with a large FOV captured in a single camera frame using 0.3 NA objective and ASI585 camera. **b,c**, Detailed look at the superficial, wing and basal epithelial layers provided by the 0.4 NA objective and ASI432 camera. **d**, Sub-basal nerve plexus with visible nerves, basal cells and dendritic cells (yellow arrows). **e**, Overlay of nerve image with NN segmentation. The full segmented FOV is provided as *Supplementary Fig. 1*. Even during short sub-second acquisition the nerves alternate their phase and their visible intensity between dark and bright (see *Supplementary Video 4*) due to the presence of the fast axial movements, known for the cornea⁴⁶. **f**, Comparison with confocal microscopy image obtained on the same subject (different corneal location). Finding the same corneal location was notably difficult with CM due to its small field-of-view and contact nature, where additional pressure on the cornea can cause the layers to curve, further reducing the useful imaging field. Consequently, the results were strongly dependent on the clinical operator's expertise.

Fig. 6. Endothelium corneal layer in health and disease. **a**, Endothelial cells of healthy subject were visible across the extended 2 mm FOV achieved using 0.3 NA objective and ASI585 camera. **b,c**, Transmission revealed the cell bodies with dark or bright interference contrast depending on their location relatively to the focal plane. **d**, Endothelium of Fuch's endothelial dystrophy patient with guttata, captured with 0.4 NA objective and ASI432 camera. **e**, Comparison with SM from the same subject, but a different (paracentral) corneal region. The central SM image was saturated due to substantial corneal haze. All presented images above are single camera frames without averaging.

Fig. 7. Corneal stroma in transmission microscopy and tomography. **a**, Transmission microscopy view of the healthy corneal stroma with visible stromal nerve. **b**, Comparative reflection image from TD-FF-OCT is more contrasted and clearly shows nuclei of keratocytes. **c**, Stroma of elderly subject had visible stripes, hypothesized to be corneal microfolds (striae). The images **a-c** were produced by averaging 15 frames. **d**, The comparative reflection TD-FF-OCT image from the same subject. Striae are highlighted using red arrows. **e**, Principle of tomographic imaging using 2 cameras. **f,g**, Comparison between transmission interference microscopy (left) and tomography (right). Yellow lines indicate the groups of keratocyte nuclei. Blue arrow points at stromal nerve. All images were captured using a 0.3 NA objective and an ASI432 camera. The images were cropped to facilitate comparison.

Fig. 8. Crystalline lens of the healthy human subject. **a**, Epithelium layer. **b**, Fiber layer with visible sutures. The images were produced using a 0.3 NA objective and by averaging approximately 40 raw frames captured with the ASI432 camera.

6. Revise the beginning of the abstract to provide a context to the work (minor).

The abstract has been adapted accordingly, while maintaining the 200-word limit.

Cellular imaging of the human anterior eye is critical for understanding complex ophthalmic diseases, yet current techniques are constrained by limited field of view or insufficient contrast. Here, we demonstrate that Ernst Abbe's foundational principles on the interference nature of transmission microscopy can be applied *in vivo* to the human eye to overcome these limitations. The transmission geometry in the eye is achieved by projecting illumination onto the posterior eye (sclera) and using the back-reflected light as a secondary illumination source for anterior eye structures. Specifically, we show that the tightly localized illumination spot at the sclera functions analogously to a closed condenser aperture in conventional microscopy, significantly enhancing interference contrast. This enables clear visualization of cells and nerves across all corneal layers within an extended 2 mm field of view. Remarkably, the crystalline lens epithelial cells, fibers, and sutures are also distinctly resolved. In patients, Fuch's endothelial dystrophy - a major ophthalmic disease affecting 300 million people - is highlighted under a new transmission contrast. Constructed using consumer-grade cameras, the instrument offers a path toward broad adoption for pre-screening and surgical follow-up, as well as for diagnosing corneal infections in low-resource settings, where anterior eye diseases are most prevalent.

The introduction has also been revised to improve clarity and language quality.

7. Line 78: Remove the word "curiously" as this effect has been previously reported by Weber and Mertz (minor).

Done:

The imaged structures in this transmission microscope appear either with dark or bright contrasts depending on their axial location relative to the optical focus (Figs. 1a-c).

8. Add a reference in line 63 (minor).

We have added the reference for the exact TD-FF-OCT device used:

Both healthy subjects and those with pathologies are examined, and the imaging results are compared with established clinical modalities (CM, SM) as well as an emerging research time-domain full-field OCT (TD-FF-OCT) device²⁹.

[29] Mazlin, V. *et al.* Compact orthogonal view OCT: clinical exploration of human trabecular meshwork and cornea at cell resolution. Preprint at <https://doi.org/10.48550/arXiv.2209.11803> (2022).

Dear Reviewers,

We are grateful for your encouraging feedback and support throughout the review process.

In this final revision, we have implemented your minor recommendations to further improve the text. Both clean and tracked versions of the revised manuscript are enclosed for your reference.

Thank you again for your valuable contributions to improving this work.

Viacheslav Mazlin
on behalf of all co-authors

Reviewer 1:

After carefully reviewing the revised manuscript, I find that all of my previous comments have been adequately addressed by the authors. Additionally, I have examined the responses to the comments raised by the other reviewers, and to the best of my knowledge, those concerns have been appropriately resolved as well. Overall, the manuscript has significantly improved. I have only one minor comment remaining, detailed below.

1. “Conversely, with a point source, the scatterer is illuminated by light of higher spatial coherence, originating from a single illumination angle.” – please consider changing to “smaller angle”.

We have adapted the text accordingly:

‘Conversely, with a point source, the scatterer is illuminated by light of higher spatial coherence, originating from a smaller illumination angle.’

Reviewer 2:

In the manuscript “Transmission interference microscopy of anterior human eye” by Alhaddad et al. present a novel application of transmission interference microscopy for in vivo imaging of the human cornea, building upon the authors’ previous work on ex vivo biological samples. The innovative use of backscattered light from the retina/sclera as a secondary illumination source is both technically elegant and practically significant. The demonstration of the role of the secondary source size in contrast modulation adds further value to the methodology. Furthermore, the implementation of this system in a clinical prototype, tested on both healthy and pathological cases, provides convincing evidence of the potential clinical relevance.

The comparison with prior literature, especially the modification of the technique described by Weber and Mertz, and the achievement of a significantly wider field of view, underscores the originality of the work. The proposed device’s affordability and portability also suggest important implications for global ophthalmic care, especially in underserved regions. Overall, the manuscript is scientifically sound, well-motivated, and clearly presented. The methods are described in sufficient detail to allow reproduction, and the conclusions are supported by the presented data.

I would like to thank the authors for their thoughtful and thorough responses to the reviewers' comments and for addressing the suggestions made in the previous round of review. In particular, I appreciate the detailed clarifications and calculations provided regarding safety considerations, which significantly enhance the credibility and completeness of the manuscript.

I only have three minor editorial suggestions before the manuscript can proceed to publication:

Line 180 – I believe the correct reference is “Fig. 4c” (not Fig. 3c).

Line 193 – It seems the authors meant “Fig. 3d” instead of “Fig. 2d).

Line 274 – Please check this reference; it likely should be “Fig. 6e” (not “Fig. 4e”).

Once these minor corrections are addressed, I believe the manuscript is ready for publication.

Indeed, we corrected the reference to figures in lines 193 and 274:

‘Based on Fig. 3d, one can deduce that this illumination is expected to provide a 3x gain in interference contrast.’

‘Moreover, the guttae exhibited dark-bright shifts across the focus further confirming that they are primarily ‘phase’ and not light absorbing objects. In comparison, reflection-based SM depicts guttae as dark spots (Fig. 6e).’

However, the reference in line 180 (Fig. 3c) is correct. The possible confusion likely stemmed from the original placement of Figure 4; we have therefore moved it closer to its mention in the text.

Reviewer 3:

The authors convincingly addressed most of the reviewers' comments. The manuscript clarity has been substantially improved as a result of these changes and is deemed for publication. I only have a few minor comments:

- When comparing TIM with existing devices, especially when imaging the central cornea of the same participant, one would expect to identify the cellular features (e.g., corneal nerves in Fig.5 and guttae in Fig. 6) from the CM/SM image on the TIM image, given its larger FOV.

We believe that the identification of similar features has not been possible so far due to the absence of a fixation target in the TIM device. Indeed, varying fixation points result in different corneal regions being imaged. Implementing a fixation target will be important for future clinical trials to ensure consistent localization.

We have added to the caption of Figure 5:

‘The corneal location does not match that of the transmission microscope, as the latter does not support precise localization within the cornea due to the absence of a fixation target.’

We have added to the discussion:

Finally, incorporating a fixation target into the transmission microscope will be essential for accurate localization of the desired corneal region and for ensuring reproducibility in clinical studies.

- It is still puzzling to me why endothelial cell boundaries are visible in the Fuchs' patient (Fig. 6d) but not in the normal participant (Fig. 6a). Further discussion would be helpful!

Indeed, we find this aspect as interesting as the reviewer does and hypothesize that it may be related not only to the presence of Fuchs' dystrophy, but also to the cataract in the same patient, which could alter the secondary

illumination incident on the cornea. We have added a sentence to the Discussion to encourage future research aimed at understanding this property of images:

'Fundamental studies of light scattering in transmission microscopy are encouraged to better understand the observed image features. For example, such studies could help explain why Fuchs' endothelial dystrophy and/or cataract enhance the visibility of endothelial cell edges, which are not apparent in healthy subjects.'